# MEMBERSHIP INFERENCE ATTACK VIA SOFT PROMPT MANIPULATION IN FEDERATED FINE-TUNING

## ABSTRACT

Membership inference attack (MIA) poses a significant privacy threat in federated learning (FL) as it allows adversaries to determine whether a client's private dataset contains a specific data sample. While defenses against membership inference attacks in standard FL have been well studied, the recent shift toward federated fine-tuning has introduced new, largely unexplored attack surfaces. To highlight this vulnerability in the emerging FL paradigm, we demonstrate that federated prompt-tuning, which adapts pre-trained models with small input prefixes to improve efficiency, also exposes a new vector for privacy attacks. We propose PROMPTMIA, a membership inference attack tailored to federated prompt-tuning, in which a malicious server can insert adversarially crafted prompts and monitors their updates during collaborative training to accurately determine whether a target data point is in a client's private dataset. We formalize this threat as a security game and empirically show that PROMPTMIA consistently attains high advantage in this game across diverse benchmark datasets. Our theoretical analysis further establishes a lower bound on the attack's advantage which explains and supports the consistently high advantage observed in our empirical results. We also investigate the effectiveness of standard membership inference defenses originally developed for gradient or output based attacks and analyze their interaction with the distinct threat landscape posed by PROMPTMIA. The results highlight non-trivial challenges for current defenses and offer insights into their limitations, underscoring the need for defense strategies that are specifically tailored to prompt-tuning in federated settings.

## 1 INTRODUCTION

While Federated Learning (FL) avoids sharing raw data, it remains vulnerable to adversarial attacks (Muñoz-González et al., 2017; Zhu et al., 2019). One prominent class of attacks is membership inference attack (MIA) (Shokri et al., 2017), where the adversarial server attempts to determine whether a specific data record was included in a client's training dataset. Most existing studies largely focus on update based attacks, which leverage information such as gradients or model parameters exchanged during training to build an attack model (Melis et al., 2019; Li et al., 2023). Recently, federated training of foundation models has shifted from full-model tuning to parameter-efficient fine-tuning (Han et al., 2024), where clients update only small subsets of parameters such as prompts (Wang et al., 2022) or adapters (Cai et al., 2022). In federated prompt tuning, clients optimize only small *soft prompt* vectors prepended to input embeddings. However, the effectiveness of MIAs in this new setting remains unclear, as the shared information is no longer the full model or gradients, but rather soft prompt vectors.

This shift in the form of shared updates introduces new challenges for existing FL based adversaries, which may no longer be applicable or effective against federated prompt tuning. At the same time, it opens up novel attack surfaces and raises important questions about the extent to which client-specific prompts can be exploited by an adversarial server to conduct privacy attacks. Previous work has examined privacy risks in LLM prompting, particularly in in-context learning (ICL), where discrete prompts (e.g., exemplars or task instructions) may leak sensitive information and have been shown vulnerable to membership inference (Wen et al., 2024; Duan et al., 2023). For instance, adversaries can craft malicious queries such as targeted questions or carefully designed input–output pairs to test whether a particular record (e.g., a patient's medical note) was included in the exemplars. To mitigate such risks, several defenses have been proposed for discrete prompts in ICL (Wu et al., 2023; Hong et al., 2023; Tang et al., 2023). In contrast, the privacy risks of soft prompts, especially in federated prompt-tuning, remain largely unexplored. More importantly, the adversarial goal differs: in ICL based MIAs, the adversary aims to determine whether a specific exemplar was included in the discrete prompt context shown to the LLM. In federated prompt-tuning, the adversary instead exploits the clients' learned prompt embeddings to infer whether a given training point was part of their local dataset.

To raise awareness of vulnerabilities in federated prompt tuning, we propose PROMPTMIA, a novel MIA specifically designed for the federated prompt tuning setting. PROMPTMIA leverages the prompt selection and update mechanism inherent to prompt learning frameworks. By injecting carefully crafted adversarial prompts into the shared prompt pool, the server can infer the presence of that sample based on whether the manipulated prompts are updated by the client. This attack achieves consistently high success rate across various benchmarks, highlighting a previously overlooked privacy vulnerability in prompt-based FL. Our contributions are as follows:

- We propose PROMPTMIA, the first MIA specifically targeting federated prompt tuning. To the best of our knowledge, this is the first MIA to exploit soft prompts as an attack surface if FL (Sec. 3).

- Unlike prior approaches that rely on gradients, model weights, or auxiliary datasets to train shadow models, PROMPTMIA exploits the prompt selection and update mechanism to infer membership in a single communication round, thereby reducing both computational cost and adversarial assumptions.

- We conduct extensive experiments across seven datasets (CIFAR-10, CIFAR-100, TinyImageNet, MNIST-M, Fashion-MNIST, CINIC-10, MMAFEDB) and three vision transformer architectures (ViT-B/32 (Dosovitskiy et al., 2020), DeiT-B/16 (Touvron et al., 2021), and ConViT (d'Ascoli et al., 2021)). Experimental results show that PROMPTMIA achieves over $90\%$ MIA success rate across diverse settings. (Sec. 4.1).

- We provide a theoretical analysis that establishes a lower bound on the advantage of PROMPTMIA, which explains and supports its strong empirical performance across diverse settings (Sec. 3.2.3).

- Since PROMPTMIA exploits the prompt selection mechanism by monitoring which prompts are selected and updated rather than the content of the update, it naturally bypasses gradient obfuscation techniques such as DPSGD-based methods (Duan et al., 2023). Moreover, we show that even when the defender is aware of of the attack, it is not trivial to mitigate PROMPTMIA using existing techniques such as outlier detection or input noise perturbation (Sections 4.2 and 4.3). Together, these results expose the limitations of current countermeasures and underscore the urgent need for research on privacy defenses specifically tailored to federated prompt-tuning.

## 2 FEDERATED PROMPT-TUNING

In this section, we explain prompt-based learning and its extension to federated prompt tuning. Prompt-based learning (Lester et al., 2021) reformulates the downstream task adaptation problem as input modification rather than weight updates. Given an image $x \in \mathbb{R}^{H \times W \times C}$ and a pretrained ViT model with frozen embedding layer $f_e$, let $x_e = f_e(x) \in \mathbb{R}^{L_x \times D}$. $L_x$ is the number of patches and , and $D$ is the dimension of the patch embeddings. Prompt tuning prepends a learnable prompt $P_e \in \mathbb{R}^{L_p \times D}$ to $x_e$ to form $x_p = [P_e; x_e]$. A frozen attention stack $f_a$ followed by classification head $f_c$ produces predictions $\hat{y} = f_c(f_a(x_p))$. Learning to Prompt (L2P) (Wang et al., 2022) maintains a prompt pool of size $M$, denoted as $\mathcal{P} = \{P_i \in \mathbb{R}^{L_p \times D}\}_{i=1}^{M}$ with corresponding learnable keys $\mathcal{K} = \{k_i \in \mathbb{R}^{D_k}\}_{i=1}^{M}$, $D_k$ is the dimension of each key vector. To facilitate query-key matching, a deterministic query function $q : \mathbb{R}^{H \times W \times C} \to \mathbb{R}^{D_k}$ is used to encode input $x$ to the same dimension as the key. We use the pretrained model as a feature extractor and define the query feature as the [CLS] representation: $q(x) = f(x)[0, :]$. Using the cosine distance function $\gamma$, or inversely the cosine similarity function $\kappa$, the top-$N$ prompts are chosen using Eq. 1:

$$\hat{\mathcal{K}}_x = \underset{\{s_i\}_{i=1}^N \subseteq [M]}{\operatorname{argmin}} \sum_{i=1}^{N} \gamma(q(x), k_{s_i}) = \underset{\{s_i\}_{i=1}^N \subseteq [M]}{\operatorname{argmax}} \sum_{i=1}^{N} \kappa(q(x), k_{s_i}), \qquad [M] = \{1, \ldots, M\}. \tag{1}$$

i.e., the $N$ prompts whose associated keys are closest to the query feature $q(x)$ under the cosine distance function $\gamma$. This is equivalent to choosing the $N$ keys with the highest cosine similarity to $q(x)$. The adapted input is $x_p = [P_{s_1}; \cdots ; P_{s_N}; x_e]$. Let the average of the prompt-token hidden states be $\bar{h}(x_p) = \operatorname{AvgPool}(f_a(x_p)[0 : NL_p, :])$ and the trainable classifier $f_c^{\phi}$ be parameterized by $\phi$, the final training objective is defined as:

$$\min_{\mathcal{P}, \mathcal{K}, \phi} \mathcal{L}(f_c^{\phi}(\bar{h}(x_p)), y) + \lambda \sum_{i \in \hat{\mathcal{K}}_x} \gamma(q(x), k_i). \tag{2}$$

The first term is the softmax cross-entropy, while the second is a surrogate that pulls the selected keys to align with their corresponding query features. In federated prompt tuning, a server keeps a global pool $\mathcal{P}_{\text{GLOBAL}} = \{P_i\}_{i=1}^{M}$. In each training round, given an input image $x$, the client selects $\hat{\mathcal{K}}_x$ using Eq. 1 and updates $\hat{\mathcal{K}}_x$, the corresponding $\hat{\mathcal{P}}_x$ along with $f_c^{\phi}$ according to Eq. 2. The server aggregates the trained prompts from participating clients to update global key-prompt pool $(\mathcal{K}_{\text{GLOBAL}}, \mathcal{P}_{\text{GLOBAL}})$ using any of the existing FL algorithms (e.g., FEDAVG (McMahan et al., 2017), FEDPROX (Li et al., 2020)) or PFPT (Weng et al., 2024). See Appendix A.1 for more details.

## 3 MIAs AGAINST FEDERATED PROMPT-TUNING

In this section, we present the formulation and workflow of PROMPTMIA. An overview of the attack is shown in Fig. 1. Rather than analyzing model gradients or outputs like previous MIA approaches, PROMPTMIA operates by injecting adversarial prompts $\mathcal{P}_{\text{ADV}}$ into the shared prompt pool $\mathcal{P}_{\text{GLOBAL}}$ prior to a training round. These prompts are associated with adversarial key vectors $\mathcal{K}_{\text{ADV}}$ tied to a target sample $\mathcal{T}$ and are designed to be activated only when $\mathcal{T}$ is present in a client's data. As clients select and update prompts using query-key matching mechanism (Eq. 1) based on their local data, the server can monitor which injected prompts are modified and use this as a membership signal. This method leverages the prompt update mechanism as a covert privacy channel, enabling accurate membership inference.

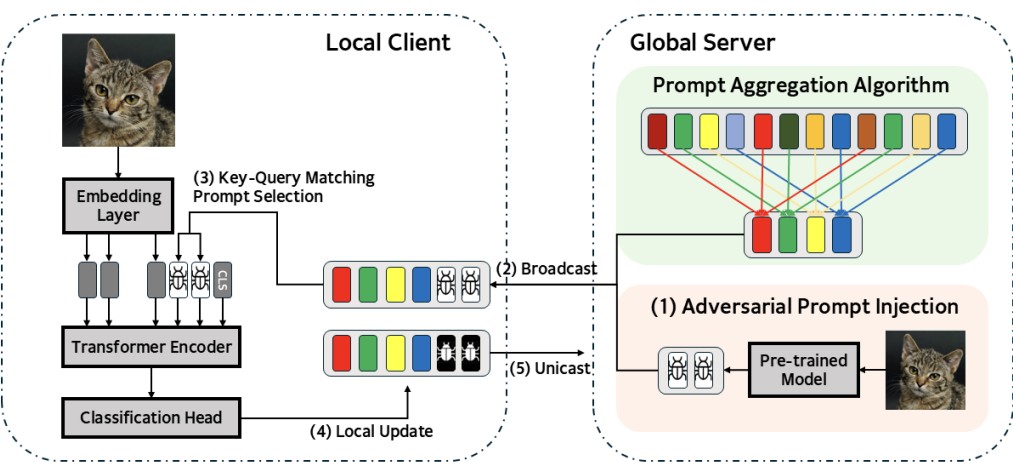

Figure 1: PROMPTMIA workflow: (1) the server injects adversarial prompts designed for a target sample into the global pool; (2) the modified pool is broadcast to clients; (3) each client performs query-key matching, which selects all adversarial prompts if the target sample is present; (4) selected prompts are locally updated and (5) returned to the server. By monitoring which prompts are updated, the server infers the target's membership in client data.

### 3.1 PROMPT BASED AMI THREAT MODELS AS SECURITY GAMES

We formalize the prompt based AMI threat models through security games between a challenger (client) and an adversary (server) in the FL setting. The adversary is denoted by $\mathcal{A}$, and the corresponding security games are represented as $\text{Exp}^{\text{AMI}}(\mathcal{A})$. In $\text{Exp}^{\text{AMI}}(\mathcal{A})$, the adversarial server $\mathcal{A}$ consists of three components: $\mathcal{A}_{\text{INIT}}$, $\mathcal{A}_{\text{ATTACK}}$, and $\mathcal{A}_{\text{GUESS}}$. At the beginning of the games, a random bit $b$ determines whether a target sample $\mathcal{T}$ belongs to the client's data $\mathcal{D}$ (with $b = 1$ indicating membership). In the initialization phase $\mathcal{A}_{\text{INIT}}$, the server constructs the current round's aggregated global key–prompt pool $(\mathcal{K}_{\text{GLOBAL}}, \mathcal{P}_{\text{GLOBAL}}) = \{(k_i, P_i)\}_{i=1}^{M}$. In the attack phase $\mathcal{A}_{\text{ATTACK}}$, the server constructs a set of $N$ adversarial keys $\mathcal{K}_{\text{ADV}} = \{k_{a_m}\}_{m=1}^{N}$, where each $k_{a_m}$ is an adversarial key vector explicitly generated for target sample $\mathcal{T}$ given query function $q$. It then selects a subset of $N$ prompts $\{(k_j, P_j)\}_{j \in S} \subseteq (\mathcal{K}_{\text{GLOBAL}}, \mathcal{P}_{\text{GLOBAL}})$ where $S \subseteq \{1, \dots, M\}$ and $|S| = N$. For each index $j \in S$, the server replaces the original key $k_j$ with an adversarial key from $\mathcal{K}_{\text{ADV}}$. While only the key values are changed, for ease of notation, we define the set of adversarial prompts as $(\mathcal{K}_{\text{ADV}}, \mathcal{P}_{\text{ADV}}) = \{(k_{a_m}, P_{a_m})\}_{m=1}^{N}$. Correspondingly, the set of remaining prompts are benign prompts $(\mathcal{K}_{\text{BENIGN}}, \mathcal{P}_{\text{BENIGN}}) = \{(k_{b_n}, P_{b_n})\}_{n=1}^{M-N}$. Together, we obtain the modified global prompt pool $(\tilde{\mathcal{K}}, \tilde{\mathcal{P}}) = (\mathcal{K}_{\text{ADV}} \cup \mathcal{K}_{\text{BENIGN}}, \mathcal{P}_{\text{ADV}} \cup \mathcal{P}_{\text{BENIGN}})$. $\mathcal{A}_{\text{ATTACK}}$ then distributes $(\tilde{\mathcal{K}}, \tilde{\mathcal{P}})$ to participating clients (instead of $(\mathcal{K}, \mathcal{P})$ like in regular federated prompt tuning). Each client $t$ selects $(\hat{\mathcal{K}}_t, \hat{\mathcal{P}}_t)$ from $(\tilde{\mathcal{K}}, \tilde{\mathcal{P}})$ using Eq. 1 and updates them based on their local data $\mathcal{D}$. In $\mathcal{A}_{\text{GUESS}}$, the server uses $\hat{\mathcal{P}}_t$ to guess $b$, effectively identifying whether $\mathcal{T} \in \mathcal{D}$. The advantage of the adversarial server $\mathcal{A}^{\mathcal{D}}$ in the security game is given by:

$$\mathbf{Adv}^{\text{AMI}}(\mathcal{A}) = \Pr[b' = 1 | b = 1] - \Pr[b' = 1 | b = 0] \tag{3}$$

where $b$ is the true membership label (1 = in dataset, 0 = not), and $b'$ is the adversary's prediction. We further define the *attack success rate (ASR)* as:

$$\mathbf{ASR}^{\text{AMI}}(\mathcal{A}) = \frac{1}{2}\left(\Pr[b' = 1 \mid b = 1] + \Pr[b' = 0 \mid b = 0]\right) = \frac{1}{2}\left(1 + \mathbf{Adv}^{\text{AMI}}(\mathcal{A})\right). \tag{4}$$

## 3.2 MEMBERSHIP INFERENCE USING ADVERSARIAL PROMPT INJECTION

Given the security game laid out in Section 3.1, the adversary's objective is to inject a set of $N$ adversarial keys $\mathcal{K}_{\text{ADV}}$ together with their corresponding prompts $\mathcal{P}_{\text{ADV}}$ into the global prompt pool such that, if the target data point $\mathcal{T} \in \mathcal{D}$, the top-$N$ selected prompts will always be the adversarial set. The membership signal is defined as the event that **all adversarial prompts are selected and subsequently updated**. To build intuition, we first introduce a *Naive Prompt Injection* attack and analyze its weaknesses, before introducing the more robust method, PROMPTMIA.

### 3.2.1 NAIVE PROMPT INJECTION ATTACK

In this naive approach, the adversary constructs adversarial keys $\mathcal{K}_{\text{ADV}} = \{\, k_{a_m} \,\}_{m=1}^{N}$ and inserts them into the global prompt pool. The goal is to ensure that if target data $\mathcal{T}$ belongs to client's dataset $\mathcal{D}$, the top-$N$ selected prompts using Eq. 1 always coincide with the adversarial set. A simple strategy is to align each adversarial key $k_{a_m}$ directly with the client's query vector $q(\mathcal{T})$ so as to maximize cosine similarity. Let $\kappa$ be the cosine similarity operator. We ensure that:

$$\kappa(q(\mathcal{T}), k_{a_m}) = (q(\mathcal{T})^{\top} k_{a_m}) \,/\, \left( \|q(\mathcal{T})\| \cdot \|k_{a_m}\| \right) = 1 \ \text{ or } \ k_{a_m} = q(\mathcal{T}), \quad \forall\, k_{a_m} \in \mathcal{K}_{\text{ADV}}. \tag{5}$$

One design choice is whether to inject new key–prompt pairs or modify existing ones. Injecting new pairs has two drawbacks: (1) clients could easily detect the sudden increase in the prompt pool size, and (2) there is no direct way to construct a matching prompt for a newly injected key. A more covert strategy is to modify existing key entries. The server selects a subset $S \subseteq [M]$ with $|S| = N$, and for each $j \in S$ performs key modificaton:

$$(k_j, P_j) \mapsto (k_{a_m}, P_j), \qquad m = 1, \dots, N, \tag{6}$$

However, this naive attack suffers from fundamental weaknesses. Since all adversarial keys collapse to the same vector $q(\mathcal{T})$ due to Eq. 5, clients can easily detect this redundancy through dimensionality reduction techniques (e.g., t-SNE, PCA) (see Fig. 2b), which would reveal a tight cluster of identical keys, or simply by directly inspecting key values. In addition, simple defenses such as discarding any key whose similarity exceeds a suspicious threshold (e.g., excluding $\kappa(q(\mathcal{T}), k_{a_m}) > 1 - r$ for small $r > 0$) can render the attack ineffective. Finally, because all adversarial keys are identical, the attack suffers from a high false positive rate: even when the target sample is absent, adversarial prompts may still be selected whenever the similarity of one adversarial key ( and as a result, all adversarial keys) to some non-target query $q(x)$, $x \in \mathcal{D}$, $x \neq \mathcal{T}$, is higher than all benign keys, or $\kappa(q(x), k_{a_m}) > \max_{k_b \in \mathcal{K}_{\text{BENIGN}}} \kappa(q(x), k_b)$.

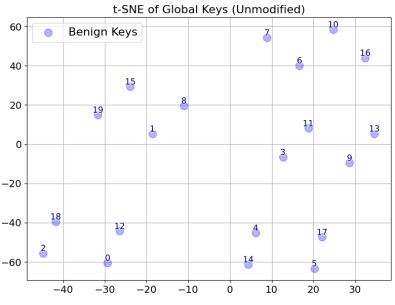 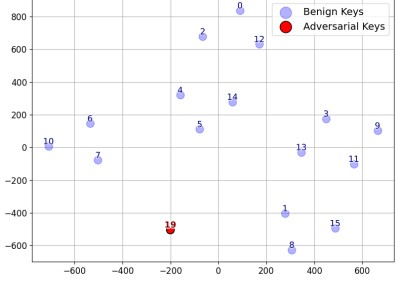 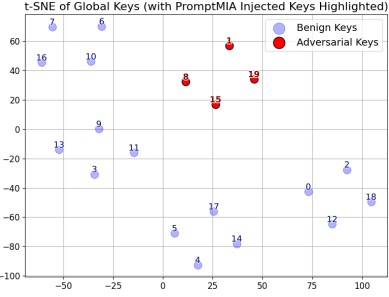

(a) t-SNE of global keys without adversarial key injection.

(b) t-SNE visualization shows naively injected adversarial keys overlapping

(c) t-SNE of global keys after PromptMIA adversarial keys injection.

Figure 2: Comparison of the global key distributions produced by a ViT-B/32 model trained on CIFAR-10 after 60 global epochs, visualized using t-SNE. Blue keys are benign keys, and red keys are adversarial keys.

### 3.2.2 PROMPTMIA

To build up on and overcome the limitations of the naive attack, we impose two requirements on adversarial key generation. First, adversarial keys must achieve higher cosine similarity with the target query $q(\mathcal{T})$ than any benign key. Second, adversarial keys must be sufficiently diverse from one another to not be trivially exposed. Formally, the server selects a subset $S \subseteq [M]$ of size $N$, and for each index $j \in S$, the original key $k_j$ is replaced with an adversarial key $k_{a_m} \in \mathcal{K}_{\text{ADV}}$ ( see Eq. 6). $k_{a_m}$ is constrained to lie within a cosine similarity interval with $q(\mathcal{T})$:

$$\kappa(q(\mathcal{T}), k_{a_m}) \in \left[ \max_{k_b \in \mathcal{K}_{\text{BENIGN}}} \kappa(q(\mathcal{T}), k_b) + \delta_{\min}, \ \max_{k_b \in \mathcal{K}_{\text{BENIGN}}} \kappa(q(\mathcal{T}), k_b) + \delta_{\min} + \Delta \right], \forall\, k_{a_m} \in \mathcal{K}_{\text{ADV}} \tag{7}$$

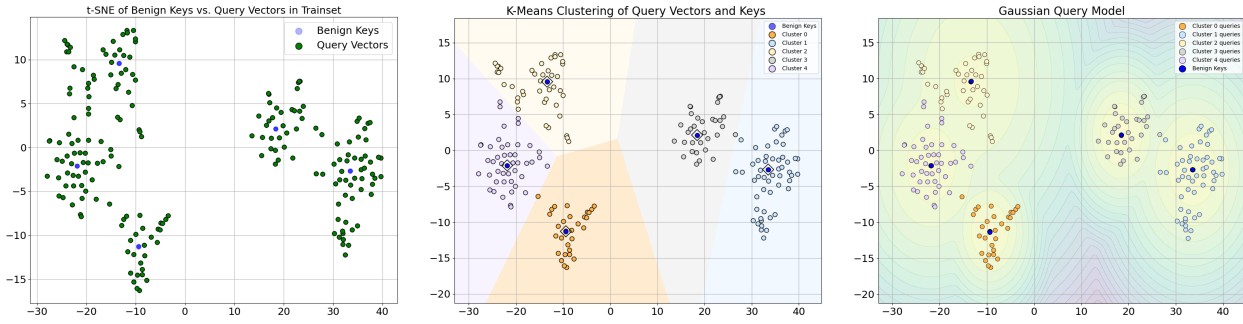

Figure 3: **Left:** t-SNE projection of the global keys and query vectors from the train set. **Center:** K-Means clustering of keys with queries. **Right:** Each cluster modeled as a spherical Gaussian distribution centered at a key. All visualizations are generated from a ViT-B32 model trained on CIFAR-10 for 60 global epochs.

where $\delta_{\min}$ ensures that $\kappa(q(\mathcal{T}), k_{a_m}) > \max_{k_b \in \mathcal{K}_{\text{BENIGN}}} \kappa(q(\mathcal{T}), k_b)$, while $\Delta$ introduces controlled variability to prevent all adversarial keys from collapsing to the same vector. Algorithm 1 describes the basic procedure for generating a single adversarial key with a specified cosine similarity $s$ to the target query. Building on this primitive, Algorithm 2 constructs a set of $N$ adversarial keys $\mathcal{K}_{\text{ADV}}$ that satisfies Eq. 7. Details of these algorithms are given in Appendix A.2. PROMPTMIA allows the adversary to generate an adversarial key set $\mathcal{K}_{\text{ADV}}$ that will always be selected and updated when $\mathcal{T} \in \mathcal{D}$, while not being easily detectable (see Fig. 2c and Section 4.2). The server generates and distributes the modified global prompt pool $(\tilde{\mathcal{K}}, \tilde{\mathcal{P}}) = (\mathcal{K}_{\text{ADV}} \cup \mathcal{K}_{\text{BENIGN}}, \ \mathcal{P}_{\text{ADV}} \cup \mathcal{P}_{\text{BENIGN}})$ to all participating clients. Given client $t$ with local dataset $\mathcal{D}$ and input data $x \in \mathcal{D}$, the client computes $q(x)$ and selects top-$N$ keys $\hat{\mathcal{K}}_x$ using Eq. 1 and update only the selected prompts using Eq. 2. If $x \equiv \mathcal{T}$, then we have the top-$N$ keys are exactly the the adversarial keys $\mathcal{K}_{\text{ADV}}$, or $\hat{\mathcal{K}}_x \equiv \mathcal{K}_{\text{ADV}}$, which corresponds precisely to the membership condition.

### 3.2.3 THEORETICAL ANALYSIS OF PROMPTMIA

In this section, we theoretically analyze the performance of PROMPTMIA. Specifically, we study the True Positive Rate (TPR), the adversary's success in identifying the target sample $\mathcal{T}$ when it is a member ($b = 1$), and the False Positive Rate (FPR) which is the adversary's error in identifying $\mathcal{T}$ when it is a non-member ($b = 0$).

First, the PROMPTMIA attack is constructed to ensure perfect identification when the target sample is present. Theorem 1 establishes that the TPR of PROMPTMIA equals 1. The proof follows directly from the construction of adversarial keys in Algorithm 2, which ensures that $\max_{k_a \in \mathcal{K}_{\text{ADV}}} \gamma(q(\mathcal{T}), k_a) < \min_{k_b \in \mathcal{K}_{\text{BENIGN}}} \gamma(q(\mathcal{T}), k_b)$. Thus, if the client processes $\mathcal{T} \in \mathcal{D}$, the top-$N$ selected keys must be exactly the set $\mathcal{K}_{\text{ADV}}$.

**Theorem 1** (True Positive Rate). *Let $\mathcal{K}_{\text{ADV}} = \{k_{a_m}\}_{m=1}^{N}$ be the set of $N$ adversarial keys generated by Algorithm 2 with parameters $\delta_{\min} > 0$ and $\Delta \geq 0$. Let $\mathcal{K}_{\text{BENIGN}}$ be the set of $M - N$ benign keys. If the client's dataset $\mathcal{D}$ contains the target sample $\mathcal{T}$ (i.e., $b = 1$) and the client's selection mechanism (Eq. 1) selects the top-$N$ prompts based on highest cosine similarity (lowest cosine distance), the set of selected keys $\hat{\mathcal{K}}_{\mathcal{T}}$ for the query $q(\mathcal{T})$ will be exactly the adversarial set: $\hat{\mathcal{K}}_{\mathcal{T}} = \mathcal{K}_{\text{ADV}}$. Consequently, the True Positive Rate (TPR) of PROMPTMIA is 1.*

$$TPR = \Pr[b' = 1 \mid b = 1] = 1$$

We now analyze the FPR, $\Pr[b' = 1 \mid b = 0]$, which is more complex. A false positive arises whenever a non-member query $q(x)$ selects all $N$ adversarial keys $\{k_{a_j}\}_{j=1}^{N}$ as its top-N choices than to any benign key $\{k_{b_i}\}_{i=1}^{M-N}$. To bound this probability, we first model the distribution of non-member query $q(x)$ relative to the benign keys. The training objective in Eq. 2 includes a surrogate loss $\gamma(q(x), k_i)$ that explicitly pulls a selected key $k_i$ to align with the query feature $q(x)$ that selected it. Thus, after several rounds of training, the benign keys $\mathcal{K}_{\text{BENIGN}}$ will stabilize and function as the centroids for the query vectors $q(x)$ generated from the clients' data (see Fig. 3 and Fig. 14). This observation forms our foundational assumptions:

**Assumption 1 (Benign Keys as Cluster Centroids).** *We assume that the $K = M - N$ benign keys $k_{b_i} \in \mathcal{K}_{\text{BENIGN}}$ act as the effective centroids of the non-member query vector distribution. Each non-member query $q(x)$ is assumed to belong to the cluster of its nearest benign key.*

Following Assumption 1, we model the distribution of non-member queries $q(x)$ belonging to benign cluster $i$ in Assumption 2. This is a reasonable assumption since the prompt learning mechanism in the federated prompt-tuning

framework models the prompt set as a sample from a Poisson point process with a Gaussian base measure and likelihood (Weng et al., 2024). The aggregated prompts serve as centroids of prompt clusters, which are optimized to lie close (on average) to different regimes of input queries in Euclidean space. Consequently, it is natural to expect the input queries to be partitioned into Gaussian clusters centered around these aggregated prompts. This clustering behavior has also been verified and visualized in Fig. 3.

**Assumption 2 (Gaussian Query Model).** *We model the distribution of non-member queries $q(x)$ belonging to benign cluster $i$ as a spherical Gaussian distribution centered at that key, i.e., $q(x) \sim \mathcal{N}(k_{b_i}, \sigma_i^2 I)$ where $\sigma_i^2$ is the variance of the non-member queries associated with that key.*

Next, we introduce the adversarial key into this setting. For a non-member query $q(x)$ drawn from cluster $i$, we define a *cluster-flip* event $E_i$ as the case when $q(x)$ selects all $N$ adversarial keys as its $N$ nearest centroids. Formally, $E_i$ is the intersection of $N \times (M - N)$ race events $A_{jl}$, where each $A_{jl}$ denotes that $q(x)$ is closer to an adversarial key $k_{a_j}$ than to a benign key $k_{b_l}$. A cluster-flip event thus arises only if the adversary wins all of these races simultaneously. The probability of the joint event $E_i$ can then be upper bounded by the probability of the single race with the lowest success probability, as formally stated in Lemma 1.

**Lemma 1.** *Let $q(x) \sim \mathcal{N}(k_{b_i}, \sigma_i^2 I)$ be a non-member query from benign cluster $i$. The probability $\Pr(E_i)$, that $q(x)$ selects all $N$ adversarial keys $\mathcal{K}_{ADV} = \{k_{a_1}, \ldots, k_{a_N}\}$ as its $N$ closest centroids, is bounded by:*

$$\Pr(E_i) \leq \min_{\substack{1 \leq j \leq N \\ 1 \leq l \leq M-N}} \Phi\left(\frac{(k_{a_j} - k_{b_l})^T k_{b_i}}{\sigma_i \|k_{a_j} - k_{b_l}\|}\right)$$

*where $\Phi(\cdot)$ is the Cumulative Distribution Function (CDF) of the standard normal distribution.*

With this lemma, we can bound the FPR. The FPR is the probability of the event $E_{FP}$ that that a single, random non-member query $q(x)$ results in a cluster-flip event. Using the Law of Total Probability, FPR can be expressed as a weighted sum of cluster-flip probabilities, where the weights correspond to the prior probabilities of clusters. Based on the bound on the probability of cluster-flip events for each cluster established in Lemma 1, we can derive a bound on the FPR which is the largest (worst-case) cluster-flip probability across all clusters, stated in Theorem 2.

**Theorem 2** (False Positive Rate). *The per-sample False Positive Rate (FPR) is bounded by:*

$$FPR = \Pr[b' = 1 \mid b = 0] \leq \max_{1 \leq i \leq M-N} \left(\min_{\substack{1 \leq j \leq N \\ 1 \leq l \leq M-N}} \Phi\left(z_{ijl}\right)\right)$$

*where $\mathcal{K}_{ADV} = \{k_{a_1}, \ldots, k_{a_N}\}$ is the set of $N$ adversarial keys, and $z_{ijl}$ is the z-score:*

$$z_{ijl} = \frac{(k_{a_j} - k_{b_l})^T k_{b_i}}{\sigma_i \|k_{a_j} - k_{b_l}\|}$$

Theorem 2 provides some insights on conditions under which the attack is most effective. First, the bound is tighter when the adversarial target $q(\mathcal{T})$ is highly distinctive. A distinctive target ensures that its corresponding adversarial keys are geometrically well separated from benign key clusters, resulting in small $\Phi(z_{ijl})$. In addition, the FPR is lower when the benign data forms tight and compact clusters in the query space around the benign keys (i.e., minimizing $\sigma_i^2$). That will decreases the likelihood that any non-member query will randomly stray into the adversary's region. Additionally, combining Theorems 1 and 2 yields the bound on the Advantage stated in Corollary 1. The detailed proofs of lemmas and theorems are provided in Appendix A.6.

**Corollary 1** (Attack Advantage). *The advantage of the adversarial server $\mathcal{A}$, which is defined as $\mathbf{Adv}^{AMI}(\mathcal{A}) = TPR - FPR$, is lower bounded by:*

$$\mathbf{Adv}^{AMI}(\mathcal{A}) \geq 1 - \max_{1 \leq i \leq M-N} \left(\min_{\substack{1 \leq j \leq N \\ 1 \leq l \leq M-N}} \Phi\left(z_{ijl}\right)\right)$$

### 3.3 MEMBERSHIP INFERENCE DEFENSES AGAINST PROMPTMIA

Although the goal of this work is not to develop new defense mechanisms, we examine how standard approaches originally designed for gradient or output based MIAs interact with PROMPTMIA. A widely used defense is **Differentially Private SGD (DPSGD)** (Abadi et al., 2016; Duan et al., 2023), which clips per-sample gradients and injects noise

before aggregation to provide $(\varepsilon, \delta)$-privacy guarantees. While DPSGD protects the content of gradients, in federated prompt tuning the top-$N$ prompt selection is independent of the gradient update procedure and therefore remains exposed, rendering DPSGD ineffective against PROMPTMIA. Therefore, we do not further evaluate DPSGD in our experiments. We consider **Input Noise Perturbation**, which adds calibrated noise directly to input pixels (Lecuyer et al., 2019). Similar to DPSGD, this approach incurs privacy-utility trade-off. Another line of defense is to use **Anomaly Detection Methods** (`IsolationForest` (Liu et al., 2008), `LocalOutlierFactor` (Breunig et al., 2000), `OneClassSVM` (Manevitz & Yousef, 2001), `SGDOneClassSVM`, and `EllipticEnvelope` (Rosseeuw, 1999) to filter out adversarial prompts. Although it would be interesting to evaluate deep learning–based anomaly detection algorithms (Do et al., 2025; Jiang et al., 2023) (e.g., autoencoders, VAEs, GAN-based models) against PROMPTMIA, we leave this for future work. Moreover, the number of prompts in the prompt pool is typically small, which raises doubts about the effectiveness of deep learning based methods in this setting. Details about these defenses are given in Appendix A.3. The effectiveness of anomaly detection and input noise perturbation against PROMPTMIA are reported in Sections 4.2 and 4.3, respectively. While our experimental results (Section 4.2) show that traditional outlier detection mechanisms fail to reliably identify adversarial keys, we take one step further and introduce a hyperparameter $\beta$ that controls the alignment between adversarial and benign keys. Increasing $\beta$ improves PROMPTMIA robustness against potential stronger anomaly detectors, albeit at the cost of reduced MIA accuracy (Appendix A.4).

## 4 EXPERIMENTAL RESULTS

We evaluate PROMPTMIA on four datasets—CIFAR-10, CIFAR-100 (Krizhevsky et al., 2009), TinyImageNet (Le & Yang, 2015), and a synthetic 4-dataset benchmark constructed by pooling MNIST-M (Lee et al., 2021), Fashion-MNIST (Xiao et al., 2017), CINIC-10 (Darlow et al., 2018), and MMAFEDB [1]—and three different models: ViT-B32, ConViT and DeiT. Experiments follow four axes: (1) measuring attack effectiveness via advantage and success rate; (2) testing robustness of PROMPTMIA against classical anomaly detection methods (Isolation Forest, LOF, One-Class SVM, Elliptic Envelope); (3) analyzing the impact of input noise perturbation defenses; and (4) conducting ablations on key hyperparameters ($M$, $N$, $\beta$, $\delta_{\min}$, $\Delta$). More information about experimental settings are given in Appx. A.5.

### 4.1 ADVANTAGE AND ATTACK SUCCESS RATE MEASUREMENT

We evaluate the performance of PROMPTMIA against Federated Prompt Tuning using two metrics: Advantage (Eq. 3) and Attack Success Rate (Eq. 4). For all experiments, we set $\delta_{\min} = 0.02$ and $\Delta = 0.05$. Unless otherwise specifically noted, we set $\beta = 0$. Following (Wang et al., 2022), the global prompt pool size is fixed at $M = 20$, and the prompt selection size at $N = 4$. In the case of batched update, given batch $B = \{(x_i, y_i)\}_{i=1}^{\ell}$, for each sample the client computes the per-sample selected key set $\hat{\mathcal{K}}_{x_i}$ and corresponding per-sample loss $\mathcal{L}_{x_i}$ (defined in Eq. 2) and updates the batch-level set of chosen keys and prompts: $\hat{\mathcal{K}}_B = \hat{\mathcal{K}}_B \bigcup \hat{\mathcal{K}}_{x_i}$ and $\hat{\mathcal{P}}_B = \hat{\mathcal{P}}_B \bigcup \mathcal{P}_{x_i}$. Batch-wise loss $\mathcal{L}_B$ is calculated by accumulating $\mathcal{L}_{x_i}$. The client update the selected keys and prompts as $\hat{\mathcal{K}}_B \leftarrow \mathcal{K}_B - \mu \nabla_{\hat{\mathcal{K}}_B} \mathcal{L}_B$ and $\hat{\mathcal{P}}_B \leftarrow \hat{\mathcal{P}}_B - \mu \nabla_{\hat{\mathcal{P}}_B} \mathcal{L}_B$, where $\mu$ is the learning rate. After receiving the client's updates, the server infers membership by checking whether all adversarial prompts are selected, i.e. $\mathbb{1}_{\{\mathcal{T} \in \mathcal{D}\}} = 1$ if $\mathcal{K}_{\text{ADV}} \subseteq \hat{\mathcal{K}}_B$, and 0 otherwise. Figure 4 shows average results of three models over four different datasets. Detailed results on individual models are given in Appendix A.7. PROMPTMIA consistently achieves near-perfect attack success rates across all models and datasets at small batch sizes, and maintains $> 90\%$ ASR and Advantage at larger batch sizes. In contrast, naive prompt injection collapses as the batch size increases, reaching Advantage of only $\approx 20\%$ against FourDataset when batch size $= 256$.

### 4.2 PERFORMANCE OF OUTLIER DETECTION METHODS AGAINST PROMPTMIA

To assess the effectiveness of outlier detection against PROMPTMIA, we frame adversarial key detection as an unsupervised anomaly-detection task over the global prompt pool. For each input $x \in \mathcal{D}$, we flip $\tilde{b} \sim \text{Bernoulli}(1/2)$: if $\tilde{b} = 1$, $N$ adversarial keys are injected into the pool $\tilde{\mathcal{K}}$; otherwise, the pool remains unmodified (clean control). We then apply `IsolationForest`, `LocalOutlierFactor`, `OneClassSVM`, `SGDOneClassSVM`, and `EllipticEnvelope` to score keys, labeling injected ones as positives. Precision and Recall are computed using ground truth and averaged across all datasets and models (detailed results in App. A.8).

Table 1, Figure 5 and Figure 12 show that naively applying anomaly detection is ineffective as a defense against PromptMIA. While IsolationForest achieves high recall, it does so by broadly flagging almost all benign keys as adversarial. LocalOutlierFactor and EllipticEnvelope detected almost none of the injected adversarial keys. OneClassSVM

---

[1] https://www.kaggle.com/datasets/yuulind/mmafedb-clean

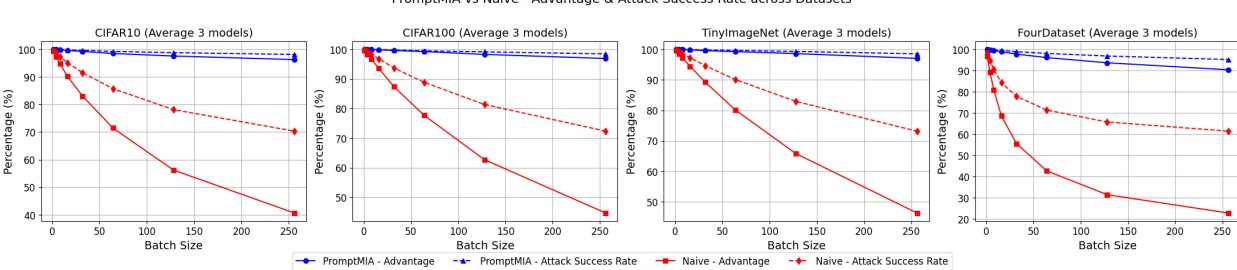

Figure 4: Performance of PromptMIA vs Naive averaged across three models. Each subplot shows Advantage and Attack Success Rate w.r.t Batch Size across CIFAR10, CIFAR100, TinyImageNet, and FourDataset.

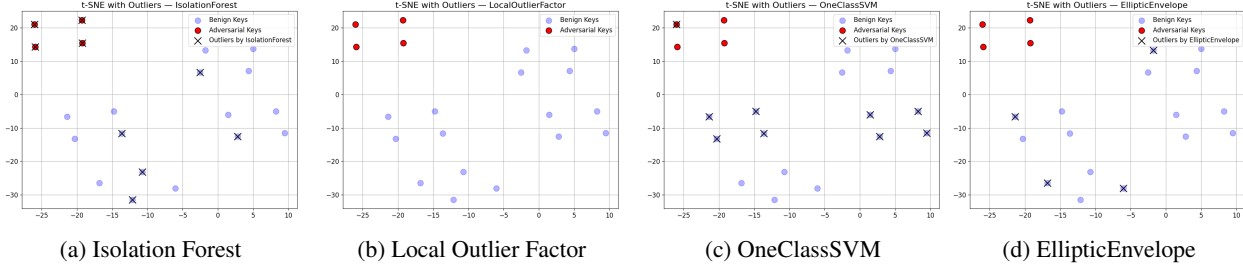

| (a) Isolation Forest | (b) Local Outlier Factor | (c) OneClassSVM | (d) EllipticEnvelope |

Figure 5: Visualization of outlier detection methods on CIFAR-10 trained Vit-B32. Blue keys are benign keys. Red keys are adversarial keys. Crossed keys are flagged as outliers from the corresponding algorithm.

Table 1: Outlier detection results averaged over 4 datasets and 3 models.

| Method | Precision | Recall | F1 |
|---|---|---|---|
| IsolationForest | **0.2672** | **1.0000** | **0.4172** |
| LocalOutlierFactor | 0.0000 | 0.0000 | 0.0000 |
| OneClassSVM | 0.2038 | 0.4993 | 0.2851 |
| EllipticEnvelope | 0.0024 | 0.0052 | 0.0033 |

Table 2: Model accuracy (%) under local differential privacy with different privacy budgets $\epsilon$ using ViT-B32.

| Dataset | $\epsilon = 3$ | $\epsilon = 5$ | $\epsilon = 8$ | Non-DP |
|---|---|---|---|---|
| CIFAR-10 | 0.85 | 0.88 | 0.90 | 0.95 |
| CIFAR-100 | 0.50 | 0.59 | 0.62 | 0.78 |
| TinyImageNet | 0.72 | 0.76 | 0.79 | 0.86 |
| FourDataset | 0.55 | 0.59 | 0.64 | 0.76 |

achieves moderate recall but also suffers from a high false positive rate. These findings highlight the ineffectiveness of traditional outlier detection methods against PROMPTMIA.

### 4.3 PERFORMANCE AND IMPACT OF NOISE PERTURBATION AGAINST PROMPTMIA

We evaluate the effectiveness of input noise perturbation as a defense against PROMPTMIA. In particular, we measure the attack success rate (ASR) of PROMPTMIA under three privacy budgets, $\epsilon \in \{3, 5, 8\}$, corresponding to high, medium, and low privacy levels, using the ViT-B32 model across four different datasets, as shown in Fig. 6. Table 2 reports the privacy–utility trade-off by presenting the model accuracy achieved under the same privacy settings.

We find that using a larger $\delta_{\min}$ (0.2–0.3) works better under input noise perturbation. If $\delta_{\min}$ is too small, the adversarial keys are only slightly closer to $q(\mathcal{T})$ than the benign keys, so even a small amount of noise can break the attack. We also observe that input noise perturbation is largely ineffective at small batch sizes across all datasets: when the batch size is 1 or 4, the ASR remains close to $95\%$ even with the strongest privacy setting ($\epsilon = 3$). As the batch size grows, increasing noise begins to reduce the attack's success, but only when very strong privacy guarantees are applied. For instance, on CIFAR-100 with $\epsilon = 3$, the ASR decreases noticeably at larger batch sizes, but this protection comes at a substantial cost in accuracy (dropping from $0.78$ to $0.50$).

At moderate privacy levels ($\epsilon = 5$ and $\epsilon = 8$), input noise fails to offer meaningful protection: the attack continues to achieve moderate to high ASR on three out of four datasets, even when the batch size is large. These results highlight a privacy–utility trade-off: strong noise (small $\epsilon$) can partially suppress PROMPTMIA but severely harms accuracy, while weaker noise (larger $\epsilon$) preserves accuracy but leaves the model more vulnerable.

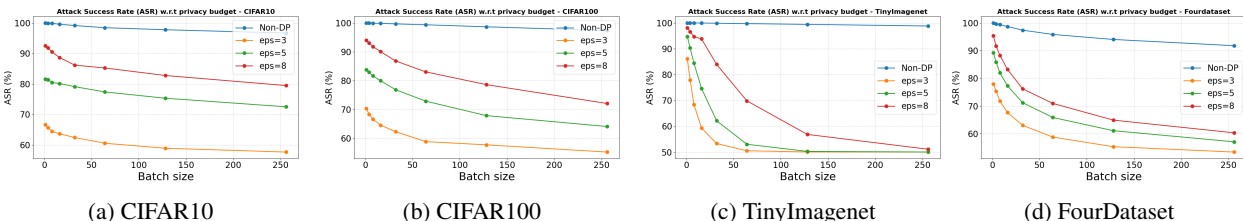

Figure 6: Attack Success Rate of PROMPTMIA under Input Noise Perturbation with different privacy budget $\epsilon$.

### 4.4 ABLATION EXPERIMENTS

We analyze how key hyperparameters (global pool size $M$, selection size $N$, $\delta_{\min}$, $\Delta$, $\beta$, and number of training rounds) affect PROMPTMIA's performance. The most interesting finding is that the attack success rate of PROMPT-MIA is much higher on the models that have been trained for a few epochs rather than randomly initialized keys (Fig. 13f). This corresponds to our theoretical findings in Section 3.2.3 that FPR is lower when the benign data forms tight and compact clusters in the query space around the benign keys, which happens naturally during the training process ( Fig. 14). More detailed analysis and insights on hyperparameters are provided in Appendix A.9. We provide additional experiments on membership information leakeage under very large batch size in Appendix A.11. To show that our attack generalizes to to all variants of federated prompt tuning that adopt the common paradigm of a frozen backbone model ( often transformer-based) paired with a shared, learnable (soft) prompt pool across clients, we provide additional experiments on multimodal and text data in Appendix A.12. Additional studies on the attack success rate and distribution of global keys and benign query vectors under heterogeneous settings is given in Appendix A.13.

### 4.5 BROADER DISCUSSION OF DEFENSE STRATEGIES

Appendix A.14 presents a detailed discussion on the implications of other potential defense strategies such as randomized key indices, secure aggregation Bonawitz et al. (2017), and prompt dropout. We show that these defenses are either provably or empirically ineffective.

## 5 RELATED WORK

**Membership Inference Attack against Federated Learning.** The goal of MIAs in FL is to identify if a specific data point was part of a client's training set. The first AMI attack in FL was introduced by (Nasr et al., 2019), which required multiple FL iterations, while (Nguyen et al., 2023) proposed a stronger single-iteration AMI attack that relied on training a separate neural network. Both approaches incur non-trivial computational costs. (Vu et al., 2024) presented two low-complexity AMI attacks that exploit fully connected and attention layers in LLMs, achieving high success rates in compromising the membership privacy of unprotected client data. More recently, Zhu et al. (2025) proposed a three-step attack that leverage updates from all clients across multiple communication rounds that can be integrated as an extension to existing attacks.

**Federated Fine-Tuning of Foundation Models.** To leverage the capabilities of large pre-trained foundation models, federated fine-tuning avoids updating the entire model and instead allows clients to fine-tune only a small subset of parameters. This approach reduces communication and computation costs while making sure raw data stay client-side. These include LoRA-based methods (Qi et al., 2024; Wang et al., 2024; Fan et al., 2025), adapter-based approaches (Cai et al., 2022; Ghiasvand et al., 2024), selective-based tuning (Zaken et al., 2021; Yu et al., 2023). Recently, prompt-based federated fine-tuning approaches have been proposed (Su et al., 2024; Weng et al., 2024; Bai et al., 2024; Feng et al., 2024), which update soft prompts instead of full model weights during FL. A soft prompt is a small set of learnable vectors prepended to the input, acting like task-specific instructions that steer model behavior. While it is possible that LoRA and adapter-based federated fine-tuning may also exhibit similar vulnerabilities, our work focuses on MIAs under federated prompt-tuning setting since these approaches are architecturally orthogonal.

**Privacy Risks in LLM Prompting.** Large language models (LLMs) are strong in-context learners that can adapt to downstream tasks by prepending discrete prompts—such as exemplars (small input–output pairs) or task instructions—without requiring fine-tuning. These exemplars, however, often contain sensitive information (e.g., medical records, proprietary text, or personally identifiable data). Adversaries can exploit this by crafting malicious prompts to extract confidential information in these discrete prompts (Wen et al., 2024; Duan et al., 2023). To mitigate such risks,

recent work has proposed a range of defense strategies primarily based on the notion of differential privacy (Duan et al., 2023; Wu et al., 2023; Hong et al., 2023; Tang et al., 2023).

## 6 CONCLUSION

In conclusion, our study reveals that federated prompt-tuning introduces a new and critical privacy vulnerability in federated learning, one that cannot be easily addressed by existing defenses designed for gradient or output based attacks. Through PROMPTMIA, we demonstrate that a malicious server can exploit adversarial prompts to reliably infer client membership, achieving consistently high attack success rate across multiple datasets. Our theoretical analysis further explains the robustness of this attack by establishing a fundamental lower bound on its advantage. Finally, our evaluation of PROMPTMIA against MIA defenses underscores their limitations in this setting, highlighting the need for new defense strategies specifically tailored to the dynamics of federated prompt tuning.

### REPODUCIBILITY STATEMENT

We are committed to ensuring the reproducibility of our work. The code for this paper, along with trained model weights, will be released upon acceptance. All datasets and pretrained weights used in our experiments are publicly available. All hyperparameters are clearly specified in the paper. We also document software dependencies and hardware used for training and evaluation.

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

How does leakage scale with large batch sizes? Can you quantify sample-level vs batch-level leakage?

# A  APPENDIX

## A.1  FEDERATED PROMPT TUNING

Figure 7 illustrates the workflow of federated prompt tuning. In federated prompt tuning, a central server maintains a global pool of prompts and keys. Each client selects the top-$N$ prompts most similar to its input features, updates those prompts and a lightweight classifier locally, and sends the updated prompts back. The server then aggregates the clients' prompts to refine the global prompt pool without sharing raw data.

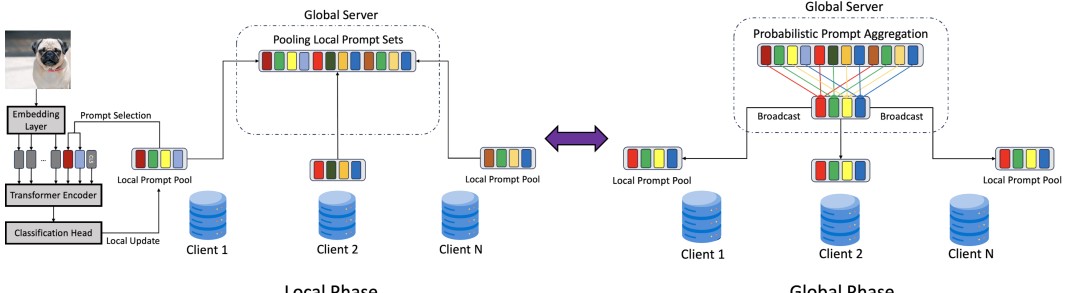

Figure 7: (Local Phase) each client samples and fine-tunes a subset of global summarizing prompts using a prompt-selection strategy; (Global Phase) the server aggregates all local prompt sets to refine the global prompt pool.

## A.2  GENERATING ADVERSARIAL KEYS SET $\mathcal{K}_{\text{ADV}}$

To mount PROMPTMIA, the adversarial server must generate a set of keys $\mathcal{K}_{\text{ADV}}$ that are more similar to the target query vector $q(\mathcal{T})$ than all benign keys $\mathcal{K}_{\text{BENIGN}}$ while remaining diverse enough to avoid detection.

**GENKEYWITHSIMILARITY** (Alg. 1) constructs a single adversarial key with a desired cosine similarity $s$ to the target query. It first *normalizes* the target query $q(\mathcal{T})$ to obtain the unit vector $\hat{q} = q(\mathcal{T})/\|q(\mathcal{T})\|$. Then it *samples a random vector* $r \in \mathbb{R}^{D_k}$ and *removes the component of $r$ that lies along $\hat{q}$* by computing $o = r - \langle r, \hat{q} \rangle \hat{q}$. This ensures that $o$ is orthogonal to the target direction. Next, $o$ is normalized to $\hat{o} = o/\|o\|$, producing a unit vector exactly perpendicular to $\hat{q}$. To construct a vector with a desired cosine similarity $s$ to $\hat{q}$, the algorithm forms

$$\hat{v} = s \cdot \hat{q} + \sqrt{1 - s^2} \cdot \hat{o},$$

Finally, $\hat{v}$ is *rescaled* to match the original norm of $q(\mathcal{T})$, producing the adversarial key $k_a = \hat{v} \cdot \|q(\mathcal{T})\|$.

**GENADVKEYSET** (Alg. 2) builds the entire adversarial key set $\mathcal{K}_{\text{ADV}}$. It computes the maximum similarity $s_{\max}$ between the target query and existing benign keys, then samples $N$ similarity scores uniformly from the interval

$$[s_{\max} + \delta_{\min}, \ s_{\max} + \delta_{\min} + \Delta],$$

ensuring that each adversarial key is slightly closer to the target query than any benign key by at least $\delta_{\min}$ but not so close that all keys collapse to the target query. Each sampled similarity $s_m$ is used to generate a key via GENKEYWITHSIMILARITY, producing a diverse set of keys that satisfy the required similarity bounds.

---

**Algorithm 1** GENKEYWITHSIMILARITY$(q(\mathcal{T}), s)$

---

**Require:** Target query vector $q(\mathcal{T}) \in \mathbb{R}^{D_k}$, desired cosine similarity $s \in (0, 1)$
**Ensure:** Adversarial key $k_a \in \mathbb{R}^{D_k}$ such that $\kappa(k_a, q(\mathcal{T})) \approx s$ and $\|k_a\| = \|q(\mathcal{T})\|$
  1: $\hat{q} \leftarrow q(\mathcal{T})/\|q(\mathcal{T})\|$                                             ▷ Normalize the target vector
  2: $r \sim \mathcal{U}(\mathbb{R}^{D_k})$                                               ▷ Sample a random vector
  3: $o \leftarrow r - \langle r, \hat{q} \rangle \cdot \hat{q}$                                      ▷ Remove component along $\hat{q}$
  4: $\hat{o} \leftarrow o/\|o\|$                                         ▷ Normalize orthogonal component
  5: $\hat{v} \leftarrow s \cdot \hat{q} + \sqrt{1 - s^2} \cdot \hat{o}$                              ▷ Combine to enforce similarity $s$
  6: $k_a \leftarrow \hat{v} \cdot \|q(\mathcal{T})\|$                                  ▷ Rescale to match original norm
  7: **return** $k_a$                                            ▷ Adversarial key

---

---

**Algorithm 2** GENADVKEYSET$(q(\mathcal{T}), \mathcal{K}_{\text{BENIGN}}, \delta_{\min}, \Delta, N)$

---

**Require:** Target query vector $q(\mathcal{T}) \in \mathbb{R}^{D_k}$, benign key set $\mathcal{K}_{\text{BENIGN}}$, margins $\delta_{\min}, \Delta$, number of adversarial keys $N$

**Ensure:** $\kappa(q(\mathcal{T}), k_{a_m}) \in \left[ \max_{k_b \in \mathcal{K}_{\text{BENIGN}}} \kappa(q(\mathcal{T}), k_b) + \delta_{\min}, \ \max_{k_b \in \mathcal{K}_{\text{BENIGN}}} \kappa(q(\mathcal{T}), k_b) + \delta_{\min} + \Delta \right], \forall k_{a_m} \in \mathcal{K}_{\text{ADV}}$

1: $\hat{q} \leftarrow q(\mathcal{T})/\|q(\mathcal{T})\|$                                      ▷ Normalize target query
2: $s_{\max} \leftarrow \max_{k_b \in \mathcal{K}_{\text{BENIGN}}} \kappa(\hat{q}, k_b)$                        ▷ Maximum similarity to benign keys
3: **for** $m = 1$ to $N$ **do**
4:      Sample $s_m \sim \mathcal{U}(s_{\max} + \delta_{\min}, \ s_{\max} + \delta_{\min} + \Delta)$
5:      $k_{a_m} \leftarrow$ GENKEYWITHSIMILARITY$(q(\mathcal{T}), s_m)$
6: **end for**
7: **return** Adversarial key set $\mathcal{K}_{\text{ADV}} = \{k_{a_m}\}_{m=1}^{N}$

---

### A.3 MEMBERSHIP INFERENCE DEFENSES

While the focus of our work is not on designing new defenses, we discuss how standard approaches—originally developed for gradient-based or output-based MIAs—can be adapted to, and interact with, PROMPTMIA.

**PromptDPSGD.** A widely used defense is *Differentially Private SGD (DPSGD)*, which clips per-sample gradients and injects noise before aggregation to provide $(\varepsilon, \delta)$-privacy guarantees. Such methods protect the *content* of gradients associated with updated prompts. However, in *federated prompt tuning*, each client still reveals which top-$N$ prompts it selects and updates, since this prompt selection mechanism is independent of the gradient update procedure. Thus, while DPSGD masks gradient values, it leaves selection patterns unchanged and thus this defense is ineffective against PROMPTMIA. DPSGD also incurs a significant privacy–utility trade-off (Abadi et al., 2016).

**Noise Perturbation** Rather than obfuscating gradient updates, clients can achieve differential privacy (DP) with respect to the input by injecting calibrated noise directly into the input image (Lecuyer et al., 2019). This introduces randomness to Eq. 1:

$$\mathcal{K}_x = \operatorname*{argmax}_{\{s_i\}_{i=1}^{N} \subseteq [M]} \sum_{i=1}^{N} \kappa\big(q(x + \eta), k_{s_i}\big), \qquad \eta \sim \mathcal{N}(0, \tilde{\sigma}I). \tag{8}$$

Increasing the noise variance strengthens the protection by making the prompt selection less predictable and reducing the effectiveness of adversarially crafted keys. However, this also comes at the high cost of model utility.

**Anomaly detection techniques** We test commonly used anomaly detection techniques in machine learning against PROMPTMIA. Specifically, we consider the following classical approaches: `IsolationForest` (Liu et al., 2008), `LocalOutlierFactor` (Breunig et al., 2000), `OneClassSVM` (Manevitz & Yousef, 2001), `SGDOneClassSVM`, and `EllipticEnvelope` (Rosseeuw, 1999). These methods are widely used, well established in the literature, and directly available in SCIKIT-LEARN library (Kramer, 2016). Although it would be interesting to evaluate deep learning–based anomaly detection algorithms (Do et al., 2025; Jiang et al., 2023) (e.g., autoencoders, VAEs, GAN-based models) against PROMPTMIA, we leave this for future work. Moreover, the number of prompts in the prompt pool is typically small (20 in our experiments), which raises doubts about the effectiveness of deep learning based methods in this setting. We therefore restrict our evaluation to the aforementioned classical methods. We briefly describe the anomaly detection algorithms use in this paper:

**Isolation Forest** detects anomalies by recursively partitioning the feature space with random splits. Each sample's path length, or the number of splits needed to isolate it in a random tree—is shorter for outliers, as they are easier to separate from the bulk of the data. Averaging this path length over a forest of random trees yields an anomaly score: points with shorter average paths are more likely to be anomalous.

**Local Outlier Factor (LOF)** detects anomalies by comparing the local density of each sample to that of its $k$-nearest neighbors. Normal points have similar density to their neighbors, while outliers lie in sparser regions. The LOF score is the ratio of the average neighbor density to the sample's own density; values $\text{LOF} > 1$ indicate potential outliers. LOF captures both local and global structure, making it effective in datasets with varying densities.

**One-Class Support Vector Machine (OCSVM)** is an unsupervised anomaly detection method derived from the Support Vector Machine framework. Instead of separating multiple classes, OCSVM learns a decision boundary that encloses the majority (normal) data in feature space, labeling points outside this region as anomalies or novelties. It works by maximizing the margin around normal data to create a robust "normalcy region" using a kernel function (commonly the radial basis function) to capture non-linear patterns.

**Elliptic Envelope** is an outlier detection method that assumes inliers follow a known distribution, typically Gaussian. It fits a robust estimate of the data's mean and covariance (using the Minimum Covariance Determinant estimator) to capture the central elliptical shape of normal data while ignoring outliers. Points are then scored by their Mahalanobis distance from this fitted ellipse, with distant points flagged as anomalies. This approach is effective when the normal data distribution is approximately Gaussian.

### A.4 CONTROLLING THE ALIGNMENT OF ADVERSARIAL AND BENIGN KEYS

While our experimental results (Section 4.2) show that it is not trivial to use traditional anomaly detection algorithms to detect adversarial prompts generated by PROMPTMIA, we also propose and extension to improve the stealthiness of PROMPTMIA in case a stronger anomaly detection algorithm is used by introducing a hyperparameter $\beta$ that controls the alignment of $\mathcal{K}_{\text{ADV}}$ and $\mathcal{K}_{\text{BENIGN}}$. In particular, we make modification to Alg. 1 as follow:

---

**Algorithm 3** GENALIGNEDKEYWITHSIMILARITY($q(\mathcal{T}), s, \mathcal{K}_{\text{BENIGN}}, \beta$)

---

**Require:** Target query vector $q(\mathcal{T}) \in \mathbb{R}^{D_k}$, desired cosine similarity $s \in (0, 1)$, benign key set $\mathcal{K}_{\text{BENIGN}}$, mixing factor $\beta \in (0, 1)$
**Ensure:** Vector $k_a \in \mathbb{R}^{D_k}$ such that $\kappa(k_a, q(\mathcal{T})) \approx s$ and $\|k_a\| = \|q(\mathcal{T})\|$
1:  $\hat{q} \leftarrow q(\mathcal{T})/\|q(\mathcal{T})\|$             ▷ Normalize target query
2:  $r \sim \mathcal{U}(\mathbb{R}^{D_k})$; $\hat{r} \leftarrow r/\|r\|$           ▷ Random unit vector
3:  Sample $k_b \sim \mathcal{K}_{\text{BENIGN}}$         ▷ Random benign key from the set
4:  $\hat{b} \leftarrow k_b/\|k_b\|$            ▷ Normalize benign key
5:  $f \leftarrow (1 - \beta) \cdot \hat{r} + \beta \cdot \hat{b}$       ▷ Mix benign key with random vector
6:  $\hat{f} \leftarrow f/\|f\|$             ▷ Normalize mixture
7:  $o \leftarrow \hat{f} - \langle \hat{f}, \hat{q} \rangle \cdot \hat{q}$       ▷ Remove component aligned with $\hat{q}$
8:  $\hat{o} \leftarrow o/\|o\|$           ▷ Normalize orthogonal component
9:  $\hat{v} \leftarrow s \cdot \hat{q} + \sqrt{1 - s^2} \cdot \hat{o}$      ▷ Construct with desired similarity $s$
10: $k_a \leftarrow \hat{v} \cdot \|q(\mathcal{T})\|$         ▷ Rescale to match target norm
11: **return** $k_a$

---

Alg. 3 extends Alg. 1 by introducing a mixing factor $\beta$ that interpolates between a random direction and a sampled benign key. By adjusting $\beta$, the adversarial key is made statistically closer to benign keys while still achieving the target cosine similarity $s$ with the query. Specifically, small values of $\beta$ result in keys that are more random and thus easier to separate from benign ones, whereas larger values of $\beta$ increase alignment with benign keys. This alignment improves the stealthiness of the attack, since anomaly detectors are more likely to misclassify adversarial keys as benign, but it also raises the false positive rate (FPR) of the attack because the top-$N$ selection mechanism may incorrectly select adversarial keys even when the target data $\mathcal{T} \notin \mathcal{D}$. Setting $\beta = 0$ reduces Alg. 3 to Alg. 1. Experimental results on the attack success rate of PROMPTMIA under various $\beta$ is given in Section A.9. Figure 8 illustrates how $\beta$ can control the alignment between adversarial and benign keys.

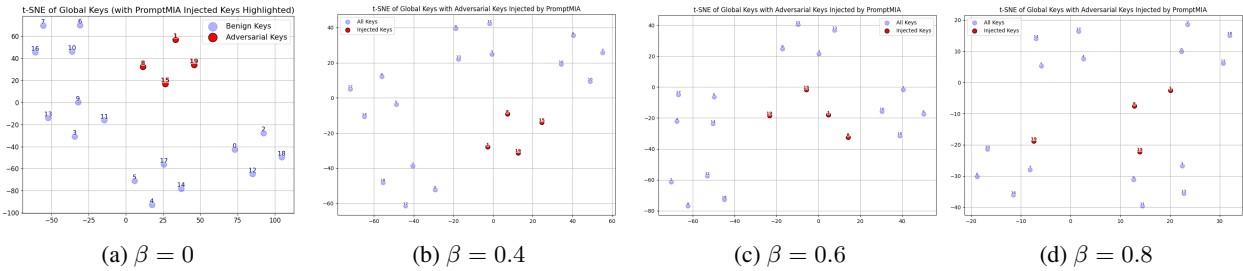

(a) $\beta = 0$     (b) $\beta = 0.4$     (c) $\beta = 0.6$     (d) $\beta = 0.8$

Figure 8: t-SNE visualization of global keys after PromptMIA injection with different $\beta$ value. Adversarial keys are colored in red, while benign keys are colored in blue.

## A.5 EXPERIMENTAL SETTINGS

**Datasets.** We evaluate our methods on four widely used vision benchmarks: CIFAR-10 and CIFAR-100 (Krizhevsky et al., 2009), TinyImageNet (Le & Yang, 2015), and a synthetic benchmark referred to as *4-dataset*. The 4-dataset benchmark is constructed by pooling four diverse datasets: 1) MNIST-M (Lee et al., 2021), 2) Fashion-MNIST (Xiao et al., 2017), 3) CINIC-10 (Darlow et al., 2018), and 4) MMAFEDB[2]. For 4-dataset, we use a total of 120,000 training and 10,000 test samples, with 30,000 training and 2,500 test examples drawn from each dataset, ensuring uniform class distributions. For CIFAR-10, CIFAR-100, and TinyImageNet, we adopt the standard train/test splits provided by the official datasets.

**Federated learning setup.** To simulate heterogeneous client data, we partition datasets using a Dirichlet distribution with concentration parameter $\alpha = 0.5$, which produces non-i.i.d. label distributions across clients. We consider a federation of 80 clients, with 10 randomly selected in each communication round. Local updates are performed with the Adam optimizer (learning rate $1 \times 10^{-4}$). Training is conducted for 60 communication rounds. We set hyperparameter $\lambda$ in eq. 2 to be 0.5, consistent with (Wang et al., 2022). We use three different baseline pretrain backbone: ViT-B/32 (Dosovitskiy et al., 2020), DeiT-B/16 (Touvron et al., 2021), and ConViT (d'Ascoli et al., 2021).

**Evaluation protocol.** We organize our evaluation along three main dimensions. First, we measure the performance of PROMPTMIA in terms of advantage ( Eq. 3) and attack success rate (Eq. 4) in Section 4.1. Second, we assess the robustness of PROMPTMIA against classical anomaly detection methods such as Isolation Forest, Local Outlier Factor, One-Class SVM, and Elliptic Envelope (Section 4.2). Third, we study the effect of noise-based defenses on the input image on PROMPTMIA attack performance (Section 4.3). Finally, we perform ablation experiments to analyze the impact of current number of training round, the role of the parameters $M$, $N$, $\beta$, $\delta_{\min}$ and $\Delta$ on the performance of PROMPTMIA (Section 4.4).

**Implementation details.** All experiments were conducted on a Linux workstation running Ubuntu 20.04 LTS, equipped with an Intel(R) Xeon(R) CPU E5-2697 v4 @ 2.30GHz (18 cores 36 threads), 384GB RAM, and two NVIDIA RTX A6000 GPU (48GB VRAM each). Our implementation is based on PyTorch 2.0 with CUDA 12.2.

## A.6 PROOFS

**Theorem 1** (True Positive Rate). *Let $\mathcal{K}_{\mathrm{ADV}} = \{k_{a_m}\}_{m=1}^N$ be the set of $N$ adversarial keys generated by Algorithm 2 with parameters $\delta_{\min} > 0$ and $\Delta \geq 0$. Let $\mathcal{K}_{\mathrm{BENIGN}}$ be the set of $M - N$ benign keys. If the client's dataset $\mathcal{D}$ contains the target sample $\mathcal{T}$ (i.e., $b = 1$) and the client's selection mechanism (Eq. 1) selects the top-$N$ prompts based on highest cosine similarity (lowest cosine distance), the set of selected keys $\mathcal{K}_{\mathcal{T}}$ for the query $q(\mathcal{T})$ will be exactly the adversarial set:*

$$\mathcal{K}_{\mathcal{T}} = \mathcal{K}_{\mathrm{ADV}}$$

*Consequently, the True Positive Rate (TPR) of PROMPTMIA is 1.*

$$TPR = \Pr[b' = 1 \mid b = 1] = 1$$

*Proof.* Based on Algorithm 2, every generated adversarial key $k_{a_m} \in \mathcal{K}_{\mathrm{ADV}}$ have a cosine similarity $s_m$ to $q(\mathcal{T})$ such that:

$$s_m \geq s_{\max} + \delta_{\min}, \qquad \text{where } s_{\max} = \max_{k_b \in \mathcal{K}_{\mathrm{BENIGN}}} \kappa(q(\mathcal{T}), k_b)$$

Since $\delta_{\min} > 0$, we have:

$$\min_{k_a \in \mathcal{K}_{\mathrm{ADV}}} \kappa(q(\mathcal{T}), k_a) > \max_{k_b \in \mathcal{K}_{\mathrm{BENIGN}}} \kappa(q(\mathcal{T}), k_b)$$

Because cosine distance $\gamma(\cdot, \cdot)$ is a monotonically decreasing function of cosine similarity $\kappa(\cdot, \cdot)$, we have an equivalent expression:

$$\max_{k_a \in \mathcal{K}_{\mathrm{ADV}}} \gamma(q(\mathcal{T}), k_a) < \min_{k_b \in \mathcal{K}_{\mathrm{BENIGN}}} \gamma(q(\mathcal{T}), k_b)$$

---

[2]https://www.kaggle.com/datasets/yuulind/mmafedb-clean

When the client processes $\mathcal{T} \in \mathcal{D}$, it computes $q(\mathcal{T})$ and selects the top-$N$ keys with the smallest distance $\gamma$ (Eq. 1). Since all $N$ adversarial keys have a smaller distance to $q(\mathcal{T})$ than all $M - N$ benign keys, the top-$N$ selected keys must be exactly the set $\mathcal{K}_{\text{ADV}}$.

The adversary's guessing rule $\mathcal{A}_{\text{GUESS}}$ predicts $b' = 1$ if all adversarial prompts $\mathcal{P}_{\text{ADV}}$ are updated. Since $b = 1$, these prompts will be selected and updated. Therefore, $\Pr[b' = 1 \mid b = 1] = 1$. $\square$

**Lemma 1** (Single-Point Flip Probability). *Let $q(x) \sim \mathcal{N}(k_{b,i}, \sigma_i^2 I)$ be a non-member query from benign cluster $i$. The probability $p_{i \to a}$ that $q(x)$ is closer to the adversarial key $k_a$ than to its own benign key $k_{b,i}$ is:*

$$p_{i \to a} = \Pr\left(\|q(x) - k_a\|^2 < \|q(x) - k_{b,i}\|^2\right) = \Phi\left(-\frac{\|k_{b,i} - k_a\|}{2\sigma_i}\right)$$

*where $\Phi(\cdot)$ is the Cumulative Distribution Function (CDF) of the standard normal distribution.*

*Proof.* The probability $p_{i \to a}$ is the probability of the "flip" event, which is defined by:
$$p_{i \to a} = \Pr\left(\|q(x) - k_a\|^2 < \|q(x) - k_{b,i}\|^2\right)$$

We first expand the squared Euclidean norms:

$$\|q(x) - k_a\|^2 < \|q(x) - k_{b,i}\|^2$$
$$\iff (q(x) - k_a)^T(q(x) - k_a) < (q(x) - k_{b,i})^T(q(x) - k_{b,i})$$
$$\iff q(x)^T q(x) - 2k_a^T q(x) + \|k_a\|^2 < q(x)^T q(x) - 2k_{b,i}^T q(x) + \|k_{b,i}\|^2$$
$$\iff -2k_a^T q(x) + \|k_a\|^2 < -2k_{b,i}^T q(x) + \|k_{b,i}\|^2$$
$$\iff 2k_{b,i}^T q(x) - 2k_a^T q(x) < \|k_{b,i}\|^2 - \|k_a\|^2$$
$$\iff 2(k_{b,i} - k_a)^T q(x) < \|k_{b,i}\|^2 - \|k_a\|^2$$

We set $Y = 2(k_{b,i} - k_a)^T q(x)$ and $C = \|k_{b,i}\|^2 - \|k_a\|^2$. Based on Assumption 2, $q(x)$ is a multivariate Gaussian $q(x) \sim \mathcal{N}(k_{b,i}, \sigma_i^2 I)$. The variable $Y$ is a linear projection of $q(x)$, so $Y$ must also follow the normal Gaussian distribution (1D). We find its mean $E[Y]$ and variance $\text{Var}(Y)$:

**Mean:**
$$E[Y] = E[(k_{b,i} - k_a)^T q(x)] = (k_{b,i} - k_a)^T E[q(x)] = (k_{b,i} - k_a)^T k_{b,i} = \|k_{b,i}\|^2 - k_a^T k_{b,i}$$

**Variance:** We have $\text{Var}(Y) = \text{Var}((k_{b,i} - k_a)^T q(x))$. Using $\text{Var}(Av) = A\text{Cov}(v)A^T$, with $A = (k_{b,i} - k_a)^T$ and $\text{Cov}(q(x)) = \sigma_i^2 I$, we can infer that:
$$\text{Var}(Y) = (k_{b,i} - k_a)^T(\sigma_i^2 I)(k_{b,i} - k_a)$$
$$\text{Var}(Y) = \sigma_i^2(k_{b,i} - k_a)^T(k_{b,i} - k_a) = \sigma_i^2\|k_{b,i} - k_a\|^2$$

Finally, we need to find $p_{i \to a} = \Pr(Y < C)$. We standardize $Y$ by converting it to $Z \sim \mathcal{N}(0, 1)$:
$$p_{i \to a} = \Pr\left(\frac{Y - E[Y]}{\sqrt{\text{Var}(Y)}} < \frac{C - E[Y]}{\sqrt{\text{Var}(Y)}}\right) = \Phi\left(\frac{C - E[Y]}{\sqrt{\text{Var}(Y)}}\right)$$

The denominator is $\sqrt{\text{Var}(Y)} = \sigma_i\|k_{b,i} - k_a\|$. The numerator is $C - E[Y]$:
$$C - E[Y] = \left[\frac{1}{2}(\|k_{b,i}\|^2 - \|k_a\|^2)\right] - \left[\|k_{b,i}\|^2 - k_a^T k_{b,i}\right]$$
$$= \frac{1}{2}\|k_{b,i}\|^2 - \frac{1}{2}\|k_a\|^2 - \|k_{b,i}\|^2 + k_a^T k_{b,i}$$
$$= -\frac{1}{2}\|k_{b,i}\|^2 + k_a^T k_{b,i} - \frac{1}{2}\|k_a\|^2$$
$$= -\frac{1}{2}\left(\|k_{b,i}\|^2 - 2k_a^T k_{b,i} + \|k_a\|^2\right)$$
$$= -\frac{1}{2}(k_{b,i} - k_a)^T(k_{b,i} - k_a)$$
$$= -\frac{1}{2}\|k_{b,i} - k_a\|^2$$

We form the $z$-score by:

$$z = \frac{-\frac{1}{2}\|k_{b,i} - k_a\|^2}{\sigma_i\|k_{b,i} - k_a\|} = -\frac{\|k_{b,i} - k_a\|}{2\sigma_i}$$

Therefore, the probability is:

$$p_{i \to a} = \Phi(z) = \Phi\left(-\frac{\|k_{b,i} - k_a\|}{2\sigma_i}\right)$$

$\square$

**Theorem 2** (FPR Bound for $N = 1$). *Let $\mathcal{B}$ be a client's batch of $B$ non-member samples which includes $b_i$ samples from each benign cluster $i$ (i.e., $\sum_i b_i = B$). The False Positive Rate (FPR) of* PROMPTMIA *with $N = 1$ is bounded by:*

$$FPR \leq \sum_{i=1}^{M-1} b_i \Phi\left(-\frac{D_{min}(i)}{2\sigma_i}\right)$$

*where $D_{min}(i) = \min_{c \in S}\|k_{b,i} - c\|$ is the minimum Euclidean distance from the benign key $k_{b,i}$ to the adversarial generation shell $S$.*

*Proof.* We aim to bound the False Positive Rate, $FPR = \Pr(b' = 1 \mid b = 0)$.

The condition $b = 0$ means the client has a batch $\mathcal{B} = \{x_1, \ldots, x_B\}$ of non-member samples. The adversary guesses $b' = 1$ if their single adversarial key $k_a$ is selected at least once. Let $E_j$ be the event that the $j$-th sample $x_j \in \mathcal{B}$ "flips", by selecting $k_a$ as its top-1 nearest neighbor:

$$E_j = \left\{\|q(x_j) - k_a\|^2 < \min_i \|q(x_j) - k_{b,i}\|^2\right\}$$

The total FPR is the probability that at least one of these events occurs (that any sample in the batch flips): $FPR = \Pr(E_1 \cup E_2 \cup \cdots \cup E_B)$.

By Boole's inequality (the Union Bound), we have:

$$FPR = \Pr\left(\bigcup_{j=1}^{B} E_j\right) \leq \sum_{j=1}^{B} \Pr(E_j)$$

We decompose this sum by the $M - 1$ benign clusters (per Assumption 1) from which the $B$ samples are drawn. We have:

$$\mu_{FP} = \sum_{j=1}^{B} \Pr(E_j) = \sum_{i=1}^{M-1} \sum_{j \text{ s.t. } x_j \in \text{cluster } i} \Pr(E_j)$$

Since all points from cluster $i$ share the same flip probability, $p_i(k_a) = \Pr(E_j \mid x_j \in \text{cluster } i)$, the sum simplifies to:

$$\mu_{FP} = \sum_{i=1}^{M-1} b_i \cdot p_i(k_a)$$

The event $E_j$ (for a point $x_j$ in cluster $i$) requires $k_a$ to be closer than all benign keys, which must include its own centroid $k_{b,i}$. Therefore, the probability $p_i(k_a)$ is bounded by the pairwise probability from Lemma 1:

$$p_i(k_a) = \Pr\left(\|q(x_j) - k_a\|^2 < \min_{i'}\|q(x_j) - k_{b,i'}\|^2\right) \leq \Pr\left(\|q(x_j) - k_a\|^2 < \|q(x_j) - k_{b,i}\|^2\right)$$

Using Lemma 1, we have:

$$p_i(k_a) \leq \Phi\left(-\frac{\|k_{b,i} - k_a\|}{2\sigma_i}\right)$$

.

Now, we find the worst case (highest probability) for this bound. The function $\Phi(-z)$ is maximized when $z$ is minimized. This occurs when the distance $\|k_{b,i} - k_a\|$ is minimized, given $k_a \in S$. We defined this as $D_{min}(i) = \min_{c \in S}\|k_{b,i} - c\|$. Thus, for *any* random $k_a \in S$, the FPR is bounded by:

$$FPR \leq \mu_{FP} = \sum_{i=1}^{M-1} b_i \cdot p_i(k_a) \leq \sum_{i=1}^{M-1} b_i \Phi\left(-\frac{D_{min}(i)}{2\sigma_i}\right)$$

$\square$

## A.7 ADVANTAGE AND ATTACK SUCCESS RATE OF INDIVIDUAL MODELS

Figures 9–11 present a detailed comparison between our proposed PROMPTMIA and a naive membership inference baseline across three backbone models (ViT-B/32, ConViT, and DeiT) on all four datasets (CIFAR10, CIFAR100, TinyImageNet, and FourDataset). Each subplot reports both the Advantage and the Attack Success Rate (ASR) as a function of the client batch size. PROMPTMIA performs worse on FourDataset compared to other dataset. This is also the dataset with the lowest predictive accuracy under no DP (see Table 2).

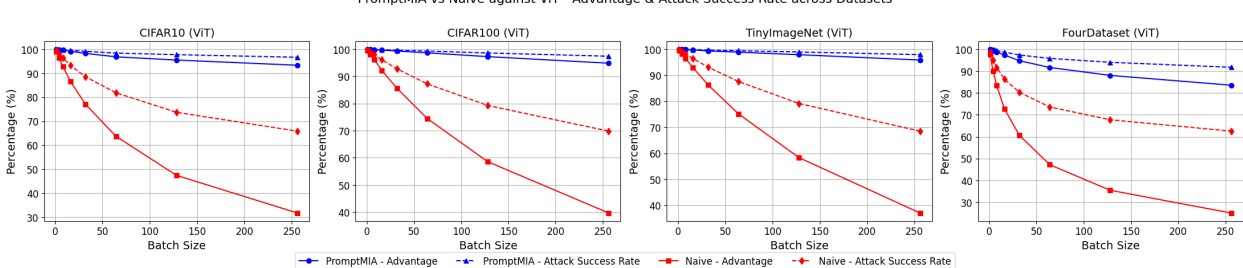

Figure 9: PromptMIA vs Naive attack results against ViT-B/32. Each subplot shows Advantage and Attack Success rate w.r.t Batch Size across CIFAR10, CIFAR100, TinyImageNet, and FourDataset.

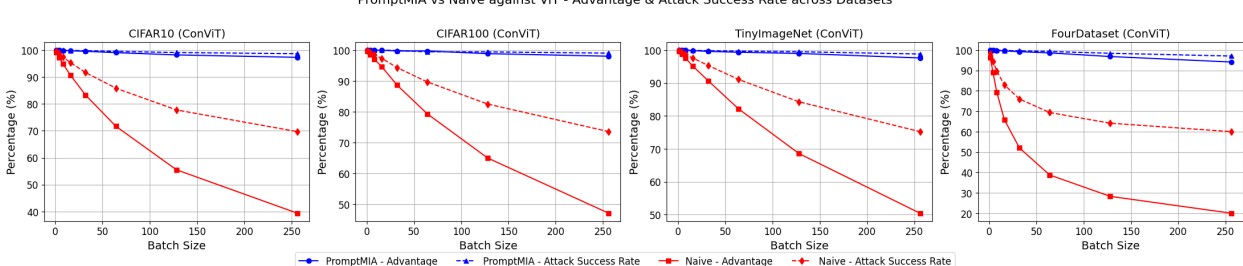

Figure 10: PromptMIA vs Naive attack results against ConViT. Each subplot shows Advantage and Attack Success rate w.r.t Batch Size across CIFAR10, CIFAR100, TinyImageNet, and FourDataset.

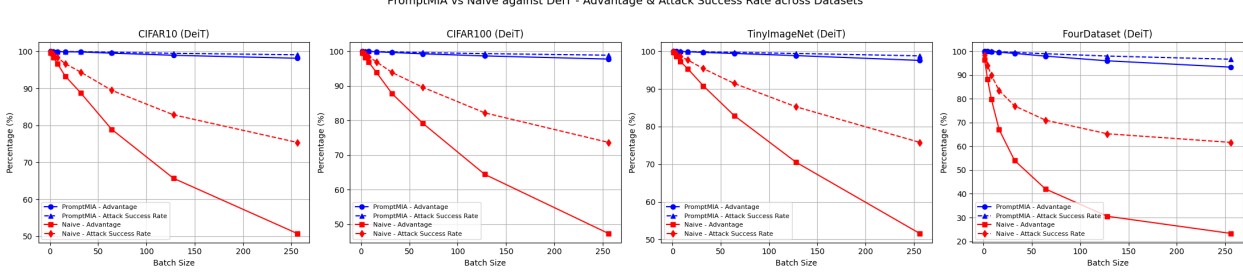

Figure 11: PromptMIA vs Naive attack results against Deit. Each subplot shows Advantage and Attack Success rate w.r.t Batch Size across CIFAR10, CIFAR100, TinyImageNet, and FourDataset.

## A.8 DETAILED RESULTS ON OUTLIER DETECTION

Table 3 reports the full precision, recall, and F1 scores of classical anomaly detection methods applied to the task of detecting adversarial keys in the global prompt pool across all datasets and backbone models. We observe that `IsolationForest` achieves the highest recall ($\approx 1.0$) on every setting, meaning it successfully flags almost all injected adversarial keys. However, its precision is low (typically 0.18–0.37), indicating that many benign keys are incorrectly labeled as adversarial, leading to high false positives. `OneClassSVM` shows slightly better precision ($\sim 0.15$–0.30) but still suffers from moderate recall and poor F1 scores. `LocalOutlierFactor` and `EllipticEnvelope` fail almost entirely in this scenario, often yielding zero detection or near-zero scores. Moreover, these methods consistently misclassify benign keys as adversarial even when no adversarial keys are present ( see Fig. 12).

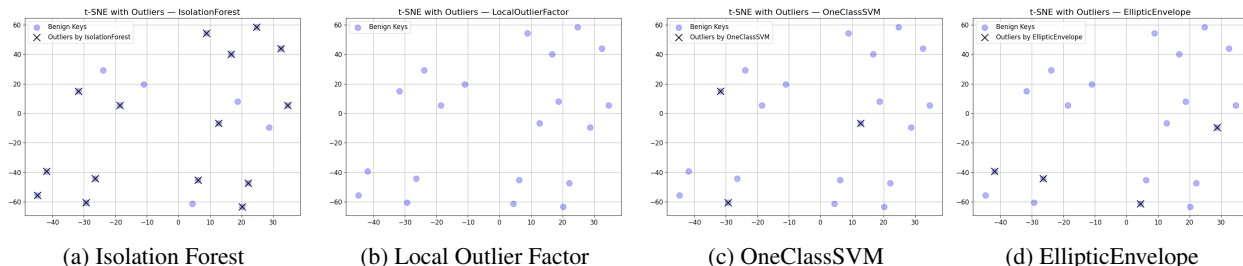

(a) Isolation Forest     (b) Local Outlier Factor     (c) OneClassSVM     (d) EllipticEnvelope

Figure 12: Visualization of outlier detection methods on CIFAR-10 trained ViT-B32. Blue keys are benign keys. Red keys are adversarial keys. Crossed keys are flagged as outliers from the corresponding algorithm. Outlier detection methods still falsely flag benign keys as outliers when no adversarial keys are present.

| Dataset | Model | Method | Precision | Recall | F1 |
|---|---|---|---|---|---|
| CIFAR10 | ViT | IsolationForest | 0.2828 | 1.0000 | 0.4409 |
| | | LocalOutlierFactor | 0.0000 | 0.0000 | 0.0000 |
| | | OneClassSVM | 0.3052 | 0.4773 | 0.3723 |
| | | EllipticEnvelope | 0.0000 | 0.0000 | 0.0000 |
| | ConViT | IsolationForest | 0.2330 | 1.0000 | 0.3779 |
| | | LocalOutlierFactor | 0.0000 | 0.0000 | 0.0000 |
| | | OneClassSVM | 0.2977 | 0.4894 | 0.3702 |
| | | EllipticEnvelope | 0.0079 | 0.0213 | 0.0116 |
| | DeiT | IsolationForest | 0.2746 | 1.0000 | 0.4308 |
| | | LocalOutlierFactor | 0.0000 | 0.0000 | 0.0000 |
| | | OneClassSVM | 0.1734 | 0.4602 | 0.2519 |
| | | EllipticEnvelope | 0.0000 | 0.0000 | 0.0000 |
| CIFAR100 | ViT | IsolationForest | 0.2576 | 1.0000 | 0.4096 |
| | | LocalOutlierFactor | 0.0000 | 0.0000 | 0.0000 |
| | | OneClassSVM | 0.2218 | 0.5196 | 0.3109 |
| | | EllipticEnvelope | 0.0000 | 0.0000 | 0.0000 |
| | DeiT | IsolationForest | 0.3623 | 1.0000 | 0.5319 |
| | | LocalOutlierFactor | 0.0000 | 0.0000 | 0.0000 |
| | | OneClassSVM | 0.2257 | 0.5450 | 0.3192 |
| | | EllipticEnvelope | 0.0022 | 0.0050 | 0.0030 |
| | ConViT | IsolationForest | 0.3128 | 1.0000 | 0.4766 |
| | | LocalOutlierFactor | 0.0000 | 0.0000 | 0.0000 |
| | | OneClassSVM | 0.1592 | 0.5045 | 0.2420 |
| | | EllipticEnvelope | 0.0000 | 0.0000 | 0.0000 |
| TinyImageNet | ViT | IsolationForest | 0.1794 | 1.0000 | 0.3042 |
| | | LocalOutlierFactor | 0.0000 | 0.0000 | 0.0000 |
| | | OneClassSVM | 0.1540 | 0.4938 | 0.2348 |
| | | EllipticEnvelope | 0.0000 | 0.0000 | 0.0000 |
| | DeiT | IsolationForest | 0.1444 | 1.0000 | 0.2524 |
| | | LocalOutlierFactor | 0.0000 | 0.0000 | 0.0000 |
| | | OneClassSVM | 0.2132 | 0.5765 | 0.3113 |
| | | EllipticEnvelope | 0.0000 | 0.0000 | 0.0000 |
| | ConViT | IsolationForest | 0.3014 | 1.0000 | 0.4632 |
| | | LocalOutlierFactor | 0.0000 | 0.0000 | 0.0000 |
| | | OneClassSVM | 0.2561 | 0.4873 | 0.3358 |
| | | EllipticEnvelope | 0.0000 | 0.0000 | 0.0000 |
| FourDataset | ViT | IsolationForest | 0.1944 | 1.0000 | 0.3255 |
| | | LocalOutlierFactor | 0.0000 | 0.0000 | 0.0000 |
| | | OneClassSVM | 0.1485 | 0.5000 | 0.2290 |
| | | EllipticEnvelope | 0.0087 | 0.0167 | 0.0114 |
| | DeiT | IsolationForest | 0.3743 | 1.0000 | 0.5447 |
| | | LocalOutlierFactor | 0.0000 | 0.0000 | 0.0000 |
| | | OneClassSVM | 0.1412 | 0.5000 | 0.2202 |
| | | EllipticEnvelope | 0.0045 | 0.0104 | 0.0062 |
| | ConViT | IsolationForest | 0.2896 | 1.0000 | 0.4492 |
| | | LocalOutlierFactor | 0.0000 | 0.0000 | 0.0000 |
| | | OneClassSVM | 0.1498 | 0.4387 | 0.2233 |
| | | EllipticEnvelope | 0.0054 | 0.0094 | 0.0069 |

Table 3: Precision, Recall, and F1 of Outlier Detection across datasets, models, and methods.

## A.9 HYPERPARAMETER ANALYSIS

To isolate the effect of each hyperparamter, all experiments in this section are conducted using Vit-B32 model and CIFAR100 dataset.

**Global Prompt Pool size $M$:** Increasing the pool size strengthens the attack. Larger $M$ (e.g., $M = 24$) yields consistently higher attack success rates across all batch sizes, while smaller pools (e.g., $M = 12$) slightly weaken the attack as batch size grows. When the pool size decreases, the probability of the adversarial keys being selected when $\mathcal{T} \notin \mathcal{D}$ increases, reducing FPR. Since the server is in control of the training protocol and the global prompt pool, they can choose the value $M$. See Fig. 13a.

**Prompt selection size $N$:** Increasing $N$ slightly weakens the attack. Again, since the server controls the training protocol, they can also most likely dictate the choice of $N$. See Fig. 13b.

**Impact of $\delta_{\min}$:** Without any defense, increasing $\delta_{\min}$ reduces the attack success rate; therefore one may choose $\delta_{\min}$ arbitrarily small (e.g. $\delta_{\min} = 0.02$). However, when the client employs input-noise perturbation, the adversary benefits from a larger $\delta_{\min}$ $(0.2 - 0.3)$, yet not so large that all adversarial keys collapse onto the target query. See Fig. 13c.

**Impact of $\Delta$:** Empirically, increasing $\Delta$ reduces inference accuracy, and a relatively small $\Delta$ does not make the attack detectable by traditional anomaly-detection methods. Therefore we choose $\Delta$ to be modest (e.g. $\Delta = 0.05$). However, $\Delta$ should not be so small that the adversarial keys become indistinguishably close to one another. See Fig. 13d.

**Impact of $\beta$:** Increasing $\beta$ increase alignment between adversarial and benign keys (see Fig. 8), but at the cost reducing attack success rate. We find $\beta = 0$ to be sufficient against traditional outlier detection methods, however carefully tuning $\beta$ might be helpful against a potentially more potent anomaly detection algorithm. See Fig. 13e.

**Impact of number of training rounds:** PROMPTMIA much higher ASR against models that have been trained for a few rounds compared to randomly initialized keys. This is reflected in our theoretical findings in Section 3.2.3 that FPR is lower when the benign data forms tight and compact clusters in the query space around the benign keys, which happens naturally during the training process. See Fig. 13f.

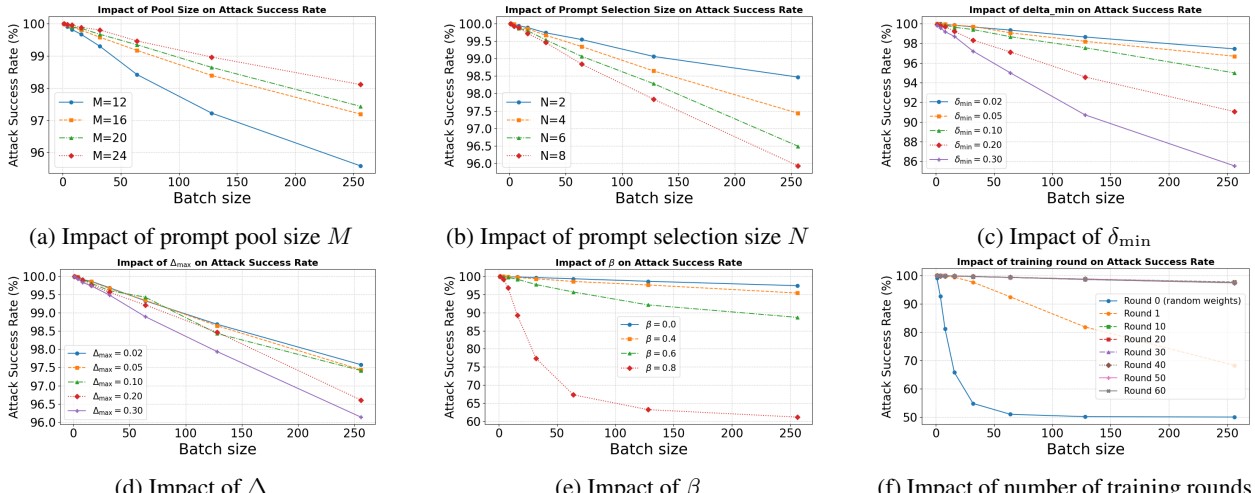

(a) Impact of prompt pool size $M$    (b) Impact of prompt selection size $N$    (c) Impact of $\delta_{\min}$

(d) Impact of $\Delta$    (e) Impact of $\beta$    (f) Impact of number of training rounds

Figure 13: Ablation study on PROMPTMIA. Each subfigure shows the effect of one parameter: (a) $M$, (b) $N$, (c) $\delta_{\min}$, (d) $\Delta$, (e) $\beta$, and (f) training rounds.

## A.10 TRAINING DYNAMICS OF BENIGN KEYS

In federated prompt tuning, the keys in the global prompt pool gradually adapt to represent the distribution of clients' data. Early in training (Round 0), benign keys are randomly initialized and do not reflect the query feature space. As training proceeds, the selected keys are pulled toward their associated query vectors, causing the benign keys to migrate toward dense regions of the feature space. Over multiple communication rounds, these keys stabilize and effectively act as cluster centroids for groups of similar queries. Figure 14 visualizes this process, showing how random keys become structured and aligned with the data distribution after the training process (Fig. 14).

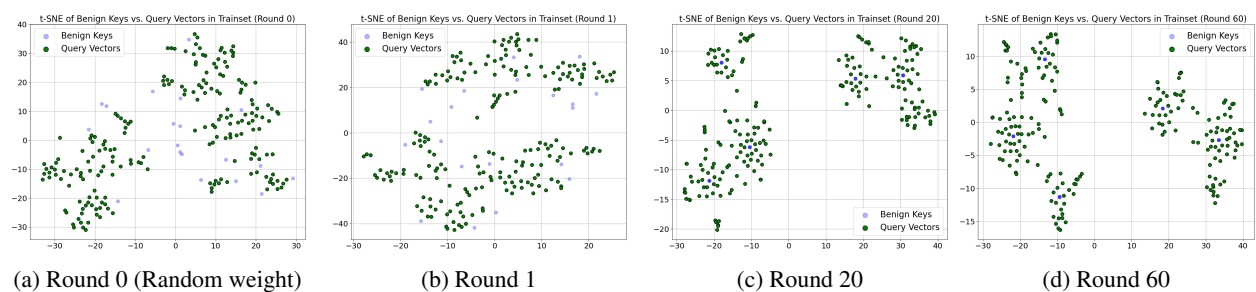

(a) Round 0 (Random weight)    (b) Round 1    (c) Round 20    (d) Round 60

Figure 14: Visualization of distribution of benign keys (blue) and query vectors $q(x)$ (green) across training rounds.

## A.11 MEMBERSHIP INFERENCE UNDER VERY LARGE BATCH SIZES

To assess whether further increasing the batch size can mitigate membership leakage, we conducted an additional set of experiments using an extreme configuration with batch size set to 1024.. As seen in Fig. 15, the attack success rate remains close to 80% on FourDataset and more than 80% on others. We also note that using such batch size is often not possible in practical scenarios where low-resource edge devices cannot afford high VRAM consumption. Defense using large batch size is therefore ineffective and impractical.

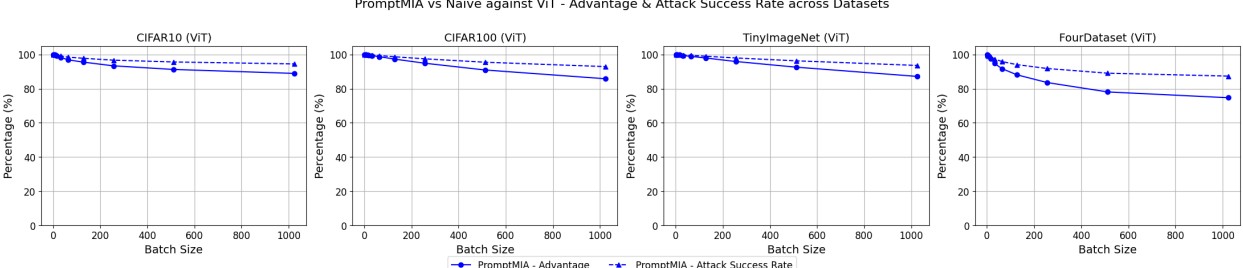

Figure 15: Attack success rate of PROMPTMIA under increasing batch size ( up to 1024) across different datasets. The results consistently show that the attack success rate remains high even under extremely large batch size.

## A.12 PERFORMANCE OF PROMPTMIA AGAINST TEXT AND MULTIMODAL DATA.

To demonstrate that PROMPTMIA extends beyond Vision Transformer, we conduct additional experiments on the UPMC Food-101 dataset Gallo et al. (2020) which is a multimodal image-text benchmark containing image–caption pairs. For these experiments, we use the pretrained Vision-and-Language Transformer (ViLT) with a frozen image encoder and frozen LLM-based text encoder (i.e., BERT). We additionally evaluate a text-only configuration by providing only textual inputs. In both the multimodal and text-only cases, PromptMIA achieves strong attack success rates, confirming that the attack is not restricted to the vision domain (see Fig. 16).

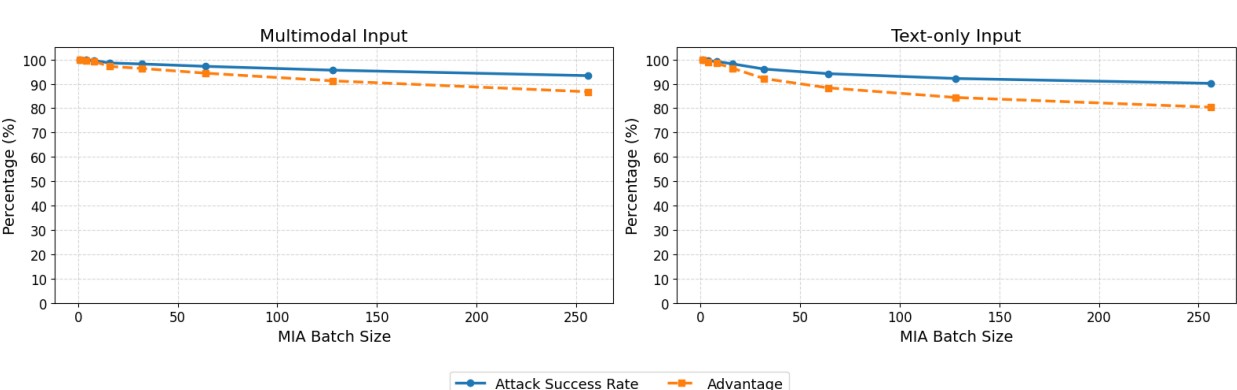

Figure 16: Attack Success Rate of PROMPTMIA under multimodal and text input modality.

## A.13 PERFORMANCE UNDER NON-HETEROGENOUS SETTINGS

In addition to Fig. 3 in the main text, we have also run additional sensitivity studies on the attack success rate under more heterogeneous setting where such assumption might be less accurate. In particular, we adopt a Dirichlet-based heterogeneous data partitioning strategy. Under this setup, each client observes samples from all classes, but the class proportions differ across clients. We generate these non-IID splits by sampling class proportions for each client from a Dirichlet($\alpha \cdot \mathbf{1}_s$) distribution over an $s$-dimensional simplex, where $s$ is the number of classes and $\alpha$ is the concentration parameter with $\alpha = 0.1$ and $\alpha = 0.5$.

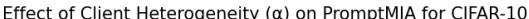

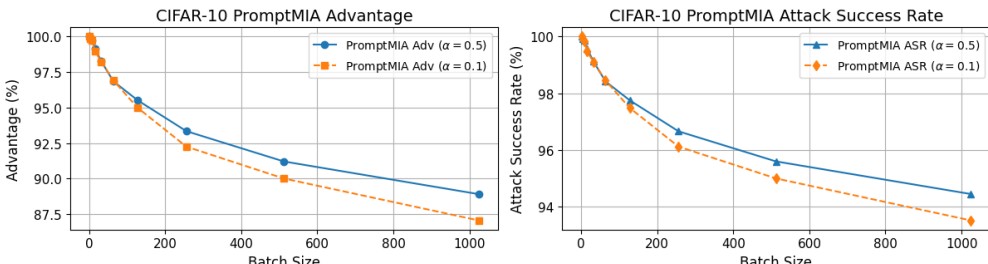

Figure 17: Attack success rate of PROMPTMIA against prompt-based FL on CIFAR-10 under different heterogeneity settings. Even on extremely large batch size, the attack success rate remains highly significant at more than 87%.

To validate **Assumption 2**, we visualize the distributions of benign keys and non-member queries for models trained on CIFAR-10 with Dirichlet parameters $\alpha = 0.1$ and $\alpha = 0.5$ below. Both plots in fact visualize empirical prompts clusters that resemble mixtures of Gaussian. Our experiments also show that the adversarial advantage and attack success rate of PromptMIA in the more extreme non-IID setting ($\alpha = 0.1$) remains significant which supports our observation above on how Assumption 2 reasonably fits the empirical prompt clusters.

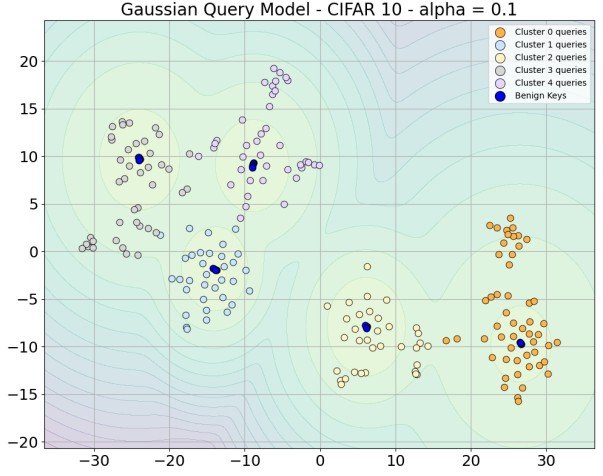

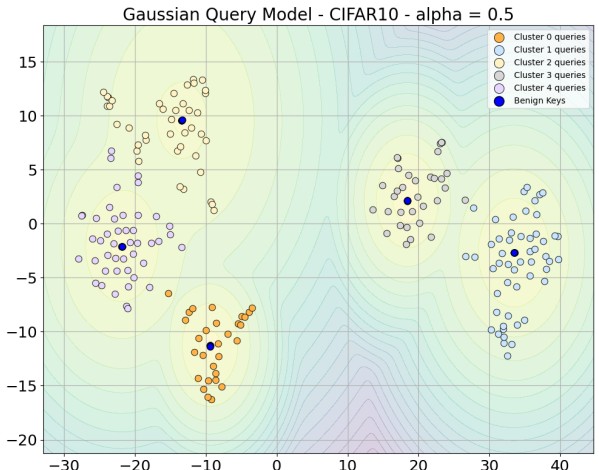

Figure 18: Visualization of prompt clusters produced by PFPT when trained on CIFAR10 with heterogeneous setting $\alpha = 0.1$

Figure 19: Visualization of prompt clusters produced by PFPT when trained on CIFAR10 with heterogeneous setting $\alpha = 0.5$

## A.14 BROADER DISCUSSION OF DEFENSE STRATEGIES

In this section, we provide in-depth discussion of other potential defense mechanisms against PROMPTMIA.

- **Randomized key indices.** We consider the setting where, after training, clients randomly permute the indices of their prompt keys before sending updates to the server to prevent the server from knowing exactly which prompts were updated. However, this defense is fundamentally ineffective against PROMPTMIA. The server already stores the previous prompt pool and can check whether each adversarial prompt appears in the client-updated pool via content matching, which is unaffected by index permutation. It then knows that the client selected and updated all adversarial prompts when no match is found. Index randomization is thus provably ineffective as it cannot prevent the server from knowing which prompts were updated.

- **Secure Aggregation.** Another strategy is to apply secure aggregation Bonawitz et al. (2017) so that individual client updates are hidden from the server. The intuition is that, if the server only observes an aggregated result rather than each client's prompt updates, it may be unable to determine whether a particular adversarial prompt was selected. However, current secure aggregation protocols are developed for linear aggregation

(that is, weighted averaging) of local model parameters, and do not extend to probablistic aggregation methods commonly used in federated prompt tuning. Weighted averaging is not sufficient in non-IID settings where local parameters must first be aligned before aggregation in order to prevent important features from collapsing into less informative representations Weng et al. (2024). To address this, the recent PFPT work Weng et al. (2024) developed an aligned and aggregated mechanism based on probabilistic non-parametric clustering which was shown to achieve substantially better performance than weighted averaging in non-IID settings. However, both the alignment and the aggregation steps in PFPT are inherently non-linear, making it unclear how existing secure aggregation methods could be generalized to support such a mechanism.

- **Prompt Dropout.** Since PROMPTMIA relies on all adversarial keys being selected, introducing randomness into the prompt selection (e.g via prompt dropout) can reduce the attack success rate but it remains substantial (larger than 75%) as shown in our experiment with prompt dropout below.

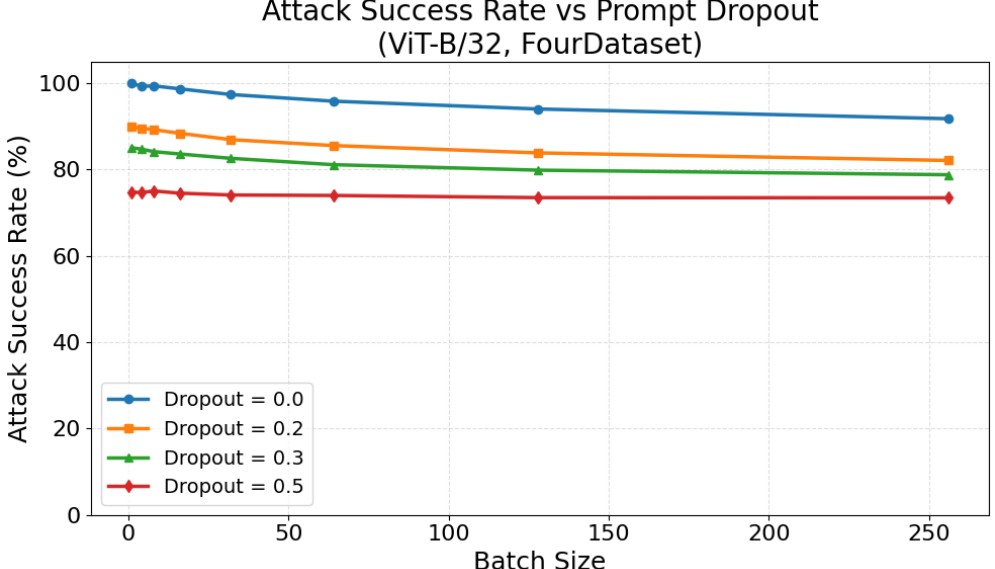

Figure 20: **Impact of Prompt Dropout on attack success rate.**

## A.15 THE USE OF LARGE LANGUAGE MODELS (LLMS)

Parts of this manuscript were edited with the assistance of a large language model (LLM) to enhance clarity, grammar, and overall readability. The LLM was used solely for language refinement; all ideas, analyses, and results remain entirely the authors' own.

