# OpenReview forum: "Membership Inference Attack via Soft Prompt Manipulation in Federated Fine-Tuning"
_ICLR.cc/2026/Conference — ICLR 2026 Conference Desk Rejected Submission_

### Official Review · Reviewer_vEpM · 2025-10-25

**Soundness:** 4
**Presentation:** 3
**Contribution:** 3
**Rating:** 8
**Confidence:** 2

**Summary:**

Parameter-efficient-fine-tuning (PEFT) used in federated learning exposes new security concerns. This paper proposes a new membership inference attack, PromptMIA, which leverages the soft prompt vectors prepended to input embeddings to explore the privacy risk associated with federated prompt tuning. PromptMIA focuses on a different attack setting where only soft prompt vectors are required by exploiting the prompt selection and update method. Specifically, PromptMIA designs the diverse adversarial keys that achieve high cosine similarity with the target query. PromptMIA then works by injecting the adversarial keys through a shared prompt pool. When a client's data includes the target key, the adversarial key is selected and updated, achieving the goal of the attack.

**Strengths:**

1. The author has made an important observation, justifying the research motivation.
2. A rigorous theoretical performance guarantee is provided for the proposed method. The paper provides carefully derived mathematical proof of the lower bounds on true positive rate(TPR) and false positive rate(FPR) of the approach.
3. Extensive studies with experiment on the existing defense mechanism deployed on federated learning to prevent membership inference attack.
4. The approach is evaluated on multiple datasets and multiple models, demonstrating the generalizability across different tasks.

**Weaknesses:**

1. The method is specifically designed on soft prompts fine tuning. There’s also other PEFT variants which also expose similar privacy concerns. I would appreciate a more detailed discussion on how the proposed method might be generalized and extend to other alternative PEFT methods.
2. In section 3.2.3, assumption 2 requires that the distribution of non-member queries q(x) belonging to a benign cluster is a spherical Gaussian distribution centered at that key. I am concerned that this is a rather strong assumption to hold outside of a controlled experimental setting, especially under non-IID client data. Sensitivity experiments or studies on how the attack and the bound response to violations of these assumptions would be worth performing.

**Questions:**

Could you respond to each weakness point?

---

> ### Author Response · Authors · 2025-11-21
>
> We thank reviewer vEpM for their positive and encouraging evaluation of our work. We appreciate their recognition that we have made ``an important observation, justifying the research motivation,`` that we provided ``a rigorous theoretical performance guarantee`` and ``extensive studies with experiments on the existing defense mechanism.``
> ***
> ``Q1``: The method is developed specifically for soft-prompt fine-tuning, but there are other PEFT approaches that may exhibit related privacy vulnerabilities. A more detailed discussion on how the proposed method could generalize or be extended to alternative PEFT techniques would further strengthen the contribution.
>
> ``A1``: We appreciate the reviewer’s suggestion. While our method is designed specifically for soft-prompt federated fine-tuning, we agree that other PEFT frameworks such as LoRA and adapter-based FL may also exhibit related vulnerabilities. These approaches, however, are architecturally orthogonal: LoRA/adapters modify internal low-rank modules, whereas prompt tuning operates entirely through input-level soft prompts and their selection mechanism, which is precisely the attack surface exploited by PROMPTMIA.
>
> Although some high-level ideas of PromptMIA might transfers to LoRA or adapter-based FL systems where clients select or activate different LoRA experts or adapters [1,2], adapting our attack to these settings would require substantial additional analysis. We therefore view MIAs for LoRA and adapter-based FL as a promising direction for future research, and we have added a brief discussion to the paper.
> ***
> [1] Zhang, Dian, and Bingyan Liu. "MoAFCL: Feature-Aware Mixture-of-Adapter for Federated Continual Learning." Proceedings of the 2025 International Conference on Multimedia Retrieval. 2025.
>
> [2] Wang, Lei, et al. "Adaptive LoRA Experts Allocation and Selection for Federated Fine-Tuning." NeurIPS 2025

---

> > ### Author Response · Authors · 2025-11-21
> >
> > ``Q2``: In Section 3.2.3, Assumption 2 requires that non-member queries \(q(x)\) from a benign cluster follow a spherical Gaussian centered at the corresponding key. While this assumption is helpful for theoretical analysis, it may be challenging to guarantee in real-world FL scenarios, especially with non-IID client data. Adding sensitivity studies or experiments that assess how the attack performance and theoretical bounds respond when this assumption does not hold would further strengthen the work.
> >
> > ``A2``: We would like to elaborate that Assumption 2 is a reasonable assumption that aligns closely with the aggregation mechanism of the PFPT model [3]. In particular,   since the prompt learning mechanism in PFPT models the prompt set as a sample from a Poisson point process with a **Gaussian base measure and likelihood** [3]. Consequently, the aggregated prompts serve as centroids of prompt clusters, which are optimized to lie close (on average) to different regimes of input queries in Euclidean space. As such, it is natural to expect the input queries to be partitioned into Gaussian clusters centered around these aggregated prompts. This clustering behavior has also been verified and visualized in Fig. 3 in the main text. Aside from this, we note that it is also established in statistics that any distribution can be approximated by a mixture of Gaussians.
> >
> > In addition to Fig. 3 in the main text, we have also conducted additional sensitivity studies on the attack success rate under more heterogeneous settings, where this assumption may be less accurate. In particular, we adopt a Dirichlet-based heterogeneous data partitioning strategy. Under this setup, each client observes samples from all classes, but the class proportions differ across clients. We generate these non-IID splits by sampling class proportions for each client from a Dirichlet$(\alpha \cdot \mathbf{1}_s)$ distribution over an $s$-dimensional simplex, where $s$ is the number of classes and $\alpha$ is the concentration parameter, with $\alpha = 0.1$ and $\alpha = 0.5$. Results can be viewed in this anonymized imgur link https://imgur.com/a/HMn7Rut or Fig. 17 in the revised manuscript.
> >
> > To validate Assumption 2, we visualize the distributions of benign keys and non-member queries for models trained on CIFAR-10 with Dirichlet parameters $\alpha = 0.1$ and $\alpha = 0.5$ at https://imgur.com/a/RUGNPcL or Fig. 18+19. Both plots in fact visualize empirical prompts clusters that resemble mixtures of Gaussian. Our experiments also show that the adversarial advantage and attack success rate of PromptMIA in the more extreme non-IID setting (\(\alpha = 0.1\)) remains significant which supports our observation above on how Assumption 2 reasonably fits the empirical prompt clusters.
> > ***
> > [3] Weng, Pei-Yau, et al. "Probabilistic federated prompt-tuning with non-IID and imbalanced data." NeurIPS 2024

---

### Official Review · Reviewer_bo1w · 2025-10-29

**Soundness:** 3
**Presentation:** 2
**Contribution:** 2
**Rating:** 4
**Confidence:** 4

**Summary:**

The paper studies membership inference in federated prompt-tuning, arguing that soft-prompt updates create a new attack surface distinct from gradient or output-based channels. It proposes PROMPTMIA, where a malicious server injects adversarial keys/prompts into the global prompt pool so that, if a target sample is present on a client, those keys are deterministically selected and updated; the server then infers membership by monitoring which prompts were updated.

**Strengths:**

1. Fresh idea. The work focuses on prompt-tuning — a new and realistic attack surface that most existing MIAs ignore. The attack setup (malicious server inserting keys) is simple but clever.

2. Theory matches intuition. The math nicely explains why the attack works better when prompts cluster tightly. It’s not just empirical; the logic behind it makes sense.

3. Solid experiments. The attack performs impressively well across several backbones and datasets, and the paper clearly shows where existing defenses fail.

**Weaknesses:**

1. Assumptions feel a bit unrealistic. The attack assumes the server can freely modify the global prompt pool and observe which prompts were updated. In many real FL systems, clients verify global models and use secure aggregation. It’d be good to discuss how the attack holds up if those protections are in place.

2. Defense part is quite weak. The tested defenses (noise, anomaly detection) are pretty basic. It would help to check more relevant defenses, like randomizing prompt selection, key rotation, or aggregation hiding which prompts were changed.

3. No comparison with recent attacks. The paper doesn’t compare against FedMIA (Zhu et al., CVPR 2025), which is a very recent and strong MIA baseline. Even a brief comparison or discussion would make the results more convincing.

4. Limited scope. All experiments are on vision transformers. It’s unclear whether this method would still work for LLM-style prompt-tuning or other PEFT methods like LoRA or adapters.

5. Unclear real-world applicability. The paper shows that PROMPTMIA works in a simulated setup, but it’s not very clear how often such server-side control or data knowledge exists in real deployments.

**Questions:**

See weakness

---

> ### Author Response · Authors · 2025-11-21
>
> We thank reviewer bo1w for their insightful feedback. We appreciate their recognition that our work introduces a ``new and realistic attack surface that most existing MIAs ignore`` and the attack setup to be ``simple but clever``. We are pleased that the reviewer found that the ``theory matches intuition`` and ``the math nicely explains why the attack works better when prompts cluster tightly``. We are further encouraged that the reviewer found our experiments solid, and ``the attack performs impressively well across several backbones and datasets, and the paper clearly shows where existing defenses fail``.
> ***
> ``Q1:`` The attack relies on the server being able to freely modify the global prompt pool and observe which prompts are updated, whereas many real-world FL deployments incorporate model verification and secure aggregation. It’d be good to discuss how the attack holds up if those protections are in place.
>
> ``A1:`` The threat model used in this paper, indeed, follows the established setting of active membership inference attacks, in which a dishonest server intentionally perturbs or poisons the global model [1,2,3]. Furthermore, in prompt-based FL under non-IID settings, clients select customized subsets of prompts for each input. As such, it is expected that there are prompts (both benign and adversarial) in the pool that are irrelevant to a client's local data. Consequently, clients do not have any signal for verifying the prompt pool and cannot detect adversarial prompts.
>
> As for secure aggregation, we note that existing protocols were developed for linear aggregation (that is, weighted averaging) of local model parameters, and do not extend to probabilistic aggregation methods commonly used in federated prompt tuning. Weighted averaging is not sufficient in non-IID settings where local parameters must first be aligned before aggregation in order to prevent important features from collapsing into less informative representations due to semantic misalignment [4]. To address this, the recent PFPT work [4] developed an aligned and aggregated mechanism based on probabilistic non-parametric clustering, which was shown to achieve substantially better performance than weighted averaging in non-IID settings. However, both the alignment and the aggregation steps in PFPT are inherently non-linear, making it unclear how existing secure aggregation methods could be generalized to support such a mechanism.
> ***
> [1] Milad Nasr, Reza Shokri, and Amir Houmansadr. Comprehensive privacy analysis of deep learning:
> Passive and active white-box inference attacks against centralized and federated learning. In 2019 IEEE
> symposium on security and privacy.
>
> [2] Truc Nguyen, et al. "Active membership inference attack under local differential privacy in federated learning". AISTATS 2023.
>
> [3] Vu, Minh, et al. "Analysis of privacy leakage in federated large language models." AISTATS 2024.
>
> [4] Weng, Pei-Yau, et al. "Probabilistic federated prompt-tuning with non-IID and imbalanced data." NeurIPS 2024

---

> ### Author Response · Authors · 2025-11-21
>
> ``Q2:`` It would help to check more relevant defenses, like randomizing prompt selection, key rotation, or aggregation hiding which prompts were changed.
>
> ``A2:`` First, we thank reviewer bo1w for highlighting several potential system-level defenses against PromptMIA. Below, we show that these defenses are either provably or empirically ineffective.
>
> **Key rotation (index permutation/randomization):** The server stores the previous prompt pool and can check whether each adversarial prompt appears in the client-updated pool via content matching, which is unaffected by index permutation (i.e., randomized key indices). It then knows that the client selected and updated all adversarial prompts when no match is found. Index randomization is thus provably ineffective as it cannot prevent the server from detecting this.
>
> **Aggregation hiding which prompts were changed:**  In federated prompt tuning, clients must share their updated prompts with the server so when an adversarial prompt is selected and updated, it can be identified via the same content matching procedure that breaks randomized key indices. While secure aggregation could in principle hide individual client updates, existing schemes do not extend to probabilistic aggregation (see our ``A1``).
>
> **Randomized prompt selection methods:** Introducing randomness into the prompt selection (e.g via prompt dropout) can reduce the attack success rate but it remains substantial (larger than 75\%) as shown in our experiment with prompt dropout in this anonymized imgur link https://imgur.com/a/csvQARF or Fig. 20 in the revised manuscript.
> ***
> ``Q3:`` No comparison with recent attacks. The paper doesn’t compare against FedMIA (Zhu et al., CVPR 2025), which is a very recent and strong MIA baseline. Even a brief comparison or discussion would make the results more convincing.
>
> ``A3:`` We would like to thank Reviewer bo1w for pointing us to the recent FedMIA work. We have cited and discussed this interesting paper in the revised manuscript. To summarize, we note that FedMIA and PROMPTMIA target fundamentally different attack surfaces. FedMIA operates in the conventional FL setting where the server receives model updates or gradients. In contrast, PROMPTMIA is the first MIA designed specifically for federated prompt tuning. This new attack surface is absent and thus complements the conventional attack surface studied in FedMIA.

---

> > ### Author Response · Authors · 2025-11-21
> >
> > ``Q4:`` It’s unclear whether this method would still work for LLM-style prompt-tuning or other PEFT methods like LoRA or adapters.
> >
> > ``A4:`` First, we would like to emphasize that our attack is task-agnostic by design and therefore applicable to all variants of federated prompt tuning that adopt the common paradigm of a frozen backbone model (often transformer-based) paired with a shared, learnable (soft) prompt pool across clients. Our attack also generalizes across input modalities and does not depend on the specifics of the pre-trained model. To demonstrate this, we conduct additional experiments on the UPMC Food-101 dataset which is a multimodal image-text benchmark containing image–caption pairs. For these experiments, we use the pretrained Vision-and-Language Transformer (ViLT) with a frozen image encoder and frozen LLM-based text encoder (i.e., BERT). We additionally evaluate a text-only configuration by providing only textual inputs. In both the multimodal and text-only cases, PromptMIA achieves strong attack success rates, confirming that the attack is not restricted to the vision domain (see https://imgur.com/a/Zkvi7tV or Fig. 16).
> >
> >  We also thank Reviewer bo1w for noting LoRA and Adapter-based FL as a promising extension. Such extension is however non-trivial and deserves a separate treatment in another follow-up paper. To elaborate, prompt-based FL relies on aggregating input adaptation (via appending prompts to the input's tokenized sequence), LoRA-based FL relies on aggregating parameter adaptation (via adding low-rank adapters to pre-trained weight matrices). Under the LoRA-based setting, the selection principle is no longer distance-based that is depended upon by our adversarial prompt design [5,6]. Designing adversarial LoRA therefore requires new innovation and significant investigation that merits a separate work.
> >
> > ***
> > ``Q5:`` The paper shows that PROMPTMIA works in a simulated setup, but it’s not very clear how often such server-side control or data knowledge exists in real deployments.
> >
> > ``A5:`` Our assumptions follow the standard FL settings, where the server receives updates from all clients and the model aggregation step occurs only on the server side. Therefore, it is realistic to assume that the server assumes control on the aggregated representation. This threat model follows the established setting of active membership inference attacks, in which a dishonest server intentionally perturbs or poisons the global model [1,2,3] Furthermore, our attack does not require any data knowledge.
> >
> > ***
> > [5] Zhang, Dian, and Bingyan Liu. "MoAFCL: Feature-Aware Mixture-of-Adapter for Federated Continual Learning." Proceedings of the 2025 International Conference on Multimedia Retrieval. 2025.
> >
> > [6] Wang, Lei, et al. "Adaptive LoRA Experts Allocation and Selection for Federated Fine-Tuning." NeurIPS 2025

---

### Official Review · Reviewer_YLTA · 2025-10-30

**Soundness:** 2
**Presentation:** 2
**Contribution:** 2
**Rating:** 4
**Confidence:** 5

**Summary:**

This paper studies membership inference in the emerging setting of federated soft prompt tuning. Instead of exploiting gradients or model parameters, the server performs a targeted prompt-key injection attack, adding adversarial keys aligned to a target sample and then inferring membership by observing which prompt keys clients select and update. The method requires no shadow models or gradient access and can succeed in a single communication round. The authors analyze detection challenges (naive injection collapses, so diversity and controlled similarity are imposed), provide a lower bound on attack advantage under a Gaussian clustering assumption, and empirically demonstrate high attack success across models and datasets. Experiments also show DP-SGD and anomaly detectors are ineffective while input noise reduces utility. The work highlights a new attack channel in federated prompt learning and argues for better defenses beyond gradient-level privacy.

**Strengths:**

1. Explores federated soft-prompt tuning and uses prompt-selection metadata instead of gradients or model parameters, representing a meaningful shift from classical MIA.

2. Works in a single round with no shadow models or gradient access and achieves consistently high ASR across models and datasets.

3. Provides a lower-bound theoretical analysis for attack advantage and validates it on multiple ViT backbones with systematic experiments.

4. Shows DP-SGD does not mitigate this threat, traditional anomaly detectors are unreliable, and input-noise defenses impose serious utility loss.

**Weaknesses:**

1. Relies on the server being able to inject prompt keys and observe prompt updates, which may not hold under secure aggregation or metadata masking in real FL systems.

2. Membership leakage under large-batch updates is not fully clarified; mapping prompt updates to individual samples may weaken in this setting.

3. Does not test realistic system-level defenses such as index randomization, prompt-dropout, DP applied to selection events, or private/local prompt pools.

4. Experiments focus on vision prompt tuning; generalization to text or multimodal prompt-tuning setups remains unclear.

**Questions:**

1. Under secure aggregation or masked updates, can the server still infer prompt indices reliably?

2. How does leakage scale with large batch sizes? Can you quantify sample-level vs batch-level leakage?

3. Can you evaluate system-level defenses such as randomized key indices, selection masking, prompt-dropout, or DP on prompt-selection?

4. Does the attack transfer to LoRA-based FL or text prompt-tuning (e.g., P-Tuning v2)?

---

> ### Author Response · Authors · 2025-11-21
>
> We thank reviewer YLTA for their thoughtful feedback. We appreciate that they recognize our work as a ``meaningful shift from classical MIA`` and our attack ``works in a single round with no shadow models or gradient access and achieves consistently high ASR across models and datasets``. We also appreciate that they recognized the value of our ``lower-bound theoretical analysis for attack advantage`` and that our study ``shows DP-SGD does not mitigate this threat, traditional anomaly detectors are unreliable, and input-noise defenses impose serious utility loss``.
>
> ***
> ``Q1:`` How does the attack behave under masking methods such as secure aggregation? Can the server still infer prompt indices reliably in these settings?
>
> ``A1:`` We would like to emphasize that current secure aggregation protocols are developed for linear aggregation (that is, weighted averaging) of local model parameters, and do not extend to probabilistic aggregation methods commonly used in federated prompt tuning. Weighted averaging is not sufficient in non-IID settings where local parameters must first be aligned before aggregation in order to prevent important features from collapsing into less informative representations due to semantic misalignment [1]. To address this, the recent PFPT work [1] developed an aligned and aggregated mechanism based on probabilistic non-parametric clustering, which was shown to achieve substantially better performance than weighted averaging in non-IID settings. However, both the alignment and the aggregation steps in PFPT are inherently non-linear, making it unclear how existing secure aggregation methods could be generalized to support such a mechanism.
>
> ***
> [1] Weng, Pei-Yau, et al. "Probabilistic federated prompt-tuning with non-IID and imbalanced data." NeurIPS 2024

---

> > ### Author Response · Authors · 2025-11-21
> >
> > ``Q2:`` How does leakage scale as batch size increases, and is it possible to quantify sample-level versus batch-level leakage?
> >
> > ``A2:``
> > **Quantify sample-level vs batch level leakage.** Our experimental results (Fig.4 and Fig.9--11 in the manuscript) already quantified sample-level versus batch-level leakage. It can be observed that increasing the training batch size does not significantly reduce the adversarial advantage (see Eq. (3)) and attack success rate. At large batch size (256) the vulnerability to MIA (i.e., data leakage) remains high with more than 85\% attack success rate.
> >
> > We have conducted a new set of experiments in an extreme cases when the batch size increases to 1024. As seen in this anonymized imgur link https://imgur.com/a/LNieMgg or Fig. 15 in the manuscript:  the attack success rate remains close to 80\% on FourDataset and more than 80\% on others. We also note that using such batch size is often not possible in practical scenarios where low-resource edge devices cannot afford high VRAM consumption. Defense using large batch size is therefore ineffective and impractical.
> >
> > **Why batch size impacts leakage.** Please refer to lines 355--364 in our manuscript for a more detailed explanation on the (slight) decrease of data leakage when batch size increases. In short, the adversarial advantage is the difference between the true and false positive rates in detecting whether a target sample belongs to a local dataset. As our attack construction guarantees that the true positive rate is $1$, the adversarial advantage decreases as the false positive rate increases. This occurs when the batch size increases, which enlarges the union set of selected prompts (each data point chooses its own subset). The enlarged union has higher chance of containing all adversarial prompts even though no individual subset contains them all, increasing the false positive rate.

---

> > > ### Author Response · Authors · 2025-11-21
> > >
> > > ``Q3:`` Can you evaluate system-level defenses such as randomized key indices, selection masking, prompt-dropout, or private/local prompt pools?
> > >
> > > ``A3:``
> > > First, we thank reviewer YLTA for highlighting several potential system-level defenses against PromptMIA. We show below that these defenses are either provably or empirically ineffective.
> > >
> > > **Randomized key indices:** The server stores the previous prompt pool and can check whether each adversarial prompt appears in the client-updated pool via content matching, which is unaffected by index permutation (i.e., randomized key indices). It then knows that the client selected and updated all adversarial prompts when no match is found. Index randomization is thus provably ineffective as it cannot prevent the server from detecting this.
> > >
> > > **Selection masking:** In federated prompt tuning, clients must share their updated prompts with the server so when an adversarial prompt is selected and updated, it can be identified via the same content matching procedure that breaks randomized key indices. While secure aggregation could in principle hide individual client updates, existing schemes do not extend to probabilistic aggregation (see ``A1``).
> > >
> > > **Private/ local prompt pools:** Federated prompt tuning requires a shared global prompt pool for cross-client aggregation, which always enables adding adversarial prompts to mount a membership inference attack (MIA).
> > >
> > > **Randomized prompt selection methods:** Introducing randomness into the prompt selection (e.g via prompt dropout) can reduce the attack success rate but it remains substantial (larger than 75\%) as shown in our experiment with prompt dropout in this anonymized imgur link https://imgur.com/a/csvQARF or Fig. 20 in the revised manuscript.

---

> > > > ### Author Response · Authors · 2025-11-21
> > > >
> > > > ``Q4:`` Does the attack generalize to text or multimodal prompt-tuning setups? Does the attack transfer to LoRA-based  FL or text prompt-tuning?
> > > >
> > > > ``A4:``
> > > >
> > > > Our attack does generalize to text or multimodal prompt-tuning setups as demonstrated below. In particular, we further evaluate PromptMIA using the multimodal UPMC Food-101 dataset [2] on an image-text classification dataset consisting of image-text pairs. We use a pretrained Vision-and-Language Transformer (ViLT) [3] which contains a frozen image encoder and a LLM-based text encoder (i.e., BERT). Our reported results in https://imgur.com/a/Zkvi7tV or Fig. 16 in the manuscript show that across both the multimodal and text-only configurations, PromptMIA achieves consistently high attack success rates, demonstrating that the attack is not limited to the vision modality.
> > > >
> > > > We appreciate that Reviewer YLTA highlighted LoRA-based FL as a potential extension. This extension is however non-trivial and deserves a separate treatment in another follow-up paper. To elaborate, prompt-based FL relies on aggregating input adaptation (via appending prompts to the input's tokenized sequence), LoRA-based FL relies on aggregating parameter adaptation (via adding low-rank adapters to pre-trained weight matrices). Under the LoRA-based setting, the selection principle is no longer distance-based that is depended upon by our adversarial prompt design [4,5]. Designing adversarial LoRA therefore requires new innovation and significant investigation that merits a separate work. Nonetheless, this does not dismiss the importance of MIA threats on prompt-based FLs which are increasingly popular due to its lightweight communication.
> > > > ***
> > > > [2] Ignazio Gallo, Gianmarco Ria, Nicola Landro, and Riccardo La Grassa. Image and text fusion for upmc food-101 using bert and cnns. In 2020 35th International Conference on Image and Vision Computing New Zealand (IVCNZ), pages 1–6. IEEE, 2020.
> > > >
> > > > [3] Wonjae Kim, Bokyung Son, and Ildoo Kim. Vilt: Vision-and-language transformer without convolution or region supervision. ICML 2021
> > > >
> > > > [4] Zhang, Dian, and Bingyan Liu. "MoAFCL: Feature-Aware Mixture-of-Adapter for Federated Continual Learning." Proceedings of the 2025 International Conference on Multimedia Retrieval. 2025.
> > > >
> > > > [5] Wang, Lei, et al. "Adaptive LoRA Experts Allocation and Selection for Federated Fine-Tuning." NeurIPS 2025

---

### Author Response · Authors · 2025-12-03

Dear ACs and Reviewers,

We sincerely appreciate the time, effort, and constructive feedback you have provided. We are encouraged by the assessments, which highlight **our contributions**:
* **Novelty**. Our work proposed a ``new and realistic attack surface`` against federated prompt tuning, representing a ``meaningful shift from classical Membership Inference Attacks`` (Reviewers bo1w \& YLTA), ``made an important observation, justifying the research motivation`` (Reviewer vEpM).
* **Strong Technical Solution.** The paper has ``theoretical performance guarantee`` for the attack’s Advantage, TPR, and FPR (all reviewers). The attack strategy has minimal overhead, ``works in a single round with no shadow models or gradient access `` (Reviewer YLTA).
* **Systematic and Comprehensive Experiments.** Our extensive evaluation demonstrates that our method ``achieves consistently high Attack Success Rate across models and datasets``, and ``clearly shows where existing defenses fail`` against our attack: DP\textendash SGD provides no meaningful protection, anomaly detectors are ineffective, and input-noise defenses significantly degrade model utility (all reviewers).

In our rebuttal, we have thoroughly **addressed all concerns** raised by the reviewers.
* **Concern 1.** How does the attack behave under system-level defenses, namely: randomized key indices, secure aggregation,  randomizing prompt selection (i.e. prompt-dropout), or private/local prompt pools (Reviewer YLTA \& bo1w)?
  * We showed that **these defenses** are either **provably** (randomized key indices, secure aggregation, private/local prompt pools), **or empirically ineffective** (prompt-dropout). We have included in-depth discussion and new experiments regarding these defenses in the rebuttal and revised manuscript ( Section A.14).

* **Concern 2.** Is the setting where the server being able to freely modify the global prompt pool and observe which prompts are updated realistic (Reviewer YLTA \&  bo1w)?
  * In the response, we provided a clarification that **our assumptions follow the standard FL settings**, and **our threat model follows the established setting of active membership inference attacks**.

* **Concern 3.** How does the leakage scale as the batch size increases to very large values ( Reviewer YLTA)?
  * We conducted **new experiments** measuring membership information leakage under **very large batch sizes** (up to 1024) across **four datasets** (CIFAR10, CIFAR100, TinyImageNet, Fourdataset). The results show that our attack continues to achieve **high success rates on all datasets**, demonstrating that **increasing batch size is an ineffective and impractical defense strategy**. We provided in-depth discussion and supporting experimental results in the rebuttal and the revised manuscript (Section~A.11).

* **Concern 4.** Does the attack generalize to text/ multimodal prompt-tuning setups ( Reviewer YLTA \& bo1w)?
  * We conducted **new experiments demonstrating** that **our attack generalizes to both text and multimodal prompt-tuning setups**. Specifically, we evaluate the attack on a pretrained Vision-and-Language Transformer (ViLT), which uses a frozen image encoder and a BERT-based text encoder, on the UPMC Food-101 dataset - a multimodal image-text benchmark containing image–caption pairs. We further evaluate a text-only configuration by providing only textual inputs. **Across both settings, PromptMIA achieves consistently high attack success rates, confirming that the attack is not limited to the vision modality**. PromptMIA achieves consistently high attack success rates, demonstrating that the attack is not limited to the vision modality.  These results are included in Section~A.12 of the revised manuscript.

* **Concern 5.** How does Assumption 2 and the attack performance holds under non-IID settings ( Reviewer vEpM)?
  * We strengthened Assumption 2 by **re-clarifying its theoretical basis** and supporting it with **new empirical evidence**. Specifically, we conducted experiments under an **extreme non-IID setting** ($\alpha = 0.1$) and analyzed both the **attack success rate** and the **distributional behavior** of benign keys and non-member queries. The corresponding results and analysis are provided in Section~A.13 of the revised manuscript.

Once again, we sincerely thank the reviewers for their constructive feedback. We have comprehensively addressed all concerns raised.

Best,

Authors

---

### Note · Program_Chairs · 2026-01-07
**Submission Desk Rejected by Program Chairs**

This paper manipulates the ICLR template to have larger margins and must be desk rejected.